# Facing Off World Model Backbones:
# RNNs, Transformers, and S4

**Fei Deng**
Rutgers University
fei.deng@rutgers.edu

**Junyeong Park**
KAIST
jyp10987@kaist.ac.kr

**Sungjin Ahn**[*]
KAIST
sungjin.ahn@kaist.ac.kr

https://fdeng18.github.io/s4wm

## Abstract

World models are a fundamental component in model-based reinforcement learning (MBRL). To perform temporally extended and consistent simulations of the future in partially observable environments, world models need to possess long-term memory. However, state-of-the-art MBRL agents, such as Dreamer, predominantly employ recurrent neural networks (RNNs) as their world model backbone, which have limited memory capacity. In this paper, we seek to explore alternative world model backbones for improving long-term memory. In particular, we investigate the effectiveness of Transformers and Structured State Space Sequence (S4) models, motivated by their remarkable ability to capture long-range dependencies in low-dimensional sequences and their complementary strengths. We propose S4WM, the first world model compatible with parallelizable SSMs including S4 and its variants. By incorporating latent variable modeling, S4WM can efficiently generate high-dimensional image sequences through latent imagination. Furthermore, we extensively compare RNN-, Transformer-, and S4-based world models across four sets of environments, which we have tailored to assess crucial memory capabilities of world models, including long-term imagination, context-dependent recall, reward prediction, and memory-based reasoning. Our findings demonstrate that S4WM outperforms Transformer-based world models in terms of long-term memory, while exhibiting greater efficiency during training and imagination. These results pave the way for the development of stronger MBRL agents.

## 1  Introduction

The human brain is frequently compared to a machine whose primary function is to construct models of the world, enabling us to predict, plan, and react to our environment effectively [51, 39]. These mental representations, referred to as world models, are integral to essential cognitive functions like decision-making and problem-solving. Similarly, one of the pivotal tasks in artificial intelligence (AI) systems that aim for human-like cognition is the development of analogous world models.

Model-Based Reinforcement Learning (MBRL) [42] has emerged as a promising approach that builds world models through interaction with the environment. As a fundamental component of MBRL, these world models empower artificial agents to anticipate the consequences of their actions and plan accordingly, leading to various advantages. Notably, MBRL offers superior sample efficiency, mitigating the high data requirements commonly associated with model-free methods. Moreover, MBRL exhibits enhanced exploration, transferability, safety, and explainability [42], making it well-suited for complex and dynamic environments where model-free methods tend to struggle.

---

[*]Correspondence to sungjin.ahn@kaist.ac.kr.

37th Conference on Neural Information Processing Systems (NeurIPS 2023).

The effectiveness and characteristics of world models crucially depend on their backbone neural network architecture. In particular, the backbone architecture dictates the model's capabilities of capturing long-term dependencies and handling stochasticity in the environment. Additionally, it affects the compactness of memory footprint and the speed of future prediction rollouts. Furthermore, in visual MBRL that finds extensive practical applications, the backbone architecture holds even greater significance than it does in state-based MBRL. This is due to the need to deal with high-dimensional, unstructured, and temporal observations.

Nevertheless, choosing the appropriate backbone architecture for visual MBRL has become a considerable challenge due to the rapidly evolving landscape of deep architectures for temporal sequence modeling. This includes the recent advancements in major backbone architecture classes, notably Transformers [61, 5] and the Structured State Space Sequence models such as S4 [21] and S5 [57].

Traditionally, Recurrent Neural Networks (RNNs) [7, 33] have been the go-to backbone architecture, thanks to their efficient use of computational resources in processing sequential data. However, RNNs tend to suffer from vanishing gradient issues [49], limiting their long-term memory capacity. Recently, Transformers [61] have demonstrated superior sequence modeling capabilities in multiple domains, including natural language processing and computer vision [12]. Their self-attention mechanism grants direct access to all previous time steps, thereby enhancing long-term memory. Moreover, Transformers offer parallel trainability and exhibit faster training speeds than RNNs. However, their quadratic complexity and slow generation speed pose challenges when dealing with very long sequences. To address these limitations, the S4 model has been proposed, offering both parallel training and recurrent generation with sub-quadratic complexity. In the Long Range Arena [59] benchmark consisting of low-dimensional sequence modeling tasks, the S4 model outperforms many Transformer variants in both task performance and computational efficiency.

In this paper, we present two primary contributions. Firstly, we introduce S4WM, the first and general world model framework that is compatible with any Parallelizable SSMs (PSSMs) including S4, S5, and other S4 variants. This is a significant development since it was unclear whether the S4 model would be effective as a high-dimensional visual world model, and if so, how this could be achieved. To this end, we instantiate the S4WM with the S4 architecture to manage high-dimensional image sequences, and propose its probabilistic latent variable modeling framework based on variational inference. Secondly, we conduct the first empirical comparative study on the three major backbone architectures for visual world modeling—RNNs, Transformers, and S4. Our results show that S4WM outperforms RNNs and Transformers across multiple memory-demanding tasks, including long-term imagination, context-dependent recall, reward prediction, and memory-based reasoning. In terms of speed, S4WM trains the fastest, while RNNs exhibit significantly higher imagination throughput. We believe that by shedding light on the strengths and weaknesses of these backbones, our study contributes to a deeper understanding that can guide researchers and practitioners in selecting suitable architectures, and potentially inspire the development of novel approaches in this field.

## 2   Related Work

**Structured State Space Sequence (S4) Model.** Originally introduced in [21], S4 is a sequence modeling framework that solves all tasks in the Long Range Arena [59] for the first time. At its core is a structured parameterization of State Space Models (SSMs) that allows efficient computation and exhibits superior performance in capturing long-range dependencies both theoretically and empirically. However, the mathematical background of S4 is quite involved. To address this, a few recent works seek to simplify, understand, and improve S4 [23, 20, 40, 22, 57, 24, 47]. It has been discovered that S4 and Transformers have complementary strengths [70, 40, 15, 34, 24]. For example, Transformers can be better at capturing local (short-range) information and performing context-dependent operations. Therefore, hybrid architectures have been proposed to achieve the best of both worlds. Furthermore, S4 and its variants have found applications in various domains, such as image and video classification [43, 36, 34, 62], audio generation [17], time-series generation [69], language modeling [70, 15, 40], and model-free reinforcement learning [10, 38]. Our study introduces the first world model compatible with S4 and its variants (more generally, parallelizable SSMs) for improving long-term memory in MBRL. We also investigate the strengths and weaknesses of S4 and Transformers in the context of world model learning.

**World Models.** World models [25] are typically implemented as dynamics models of the environment that enable the agent to plan into the future and learn policies from imagined trajectories. RNNs have been the predominant backbone architecture of world models. A notable example is RSSM [28], which has been widely used in both reconstruction-based [29–31, 65, 56, 16, 35, 63, 64, 68] and reconstruction-free [44–46, 11, 26] MBRL agents. With the advent of Transformers [61], recent works have also explored using Transformers as the world model backbone [5, 41, 54]. While Transformers are less prone to vanishing gradients [49] than RNNs, their quadratic complexity limits their applicability to long sequences. For example, recent works [41, 54] use a short imagination horizon of ~20 steps. In contrast, S4WM can successfully imagine hundreds of steps into the future with sub-quadratic complexity. We also develop an improved Transformer-based world model that can deal with long sequences by employing Transformer-XL [9].

**Agent Memory Benchmarks.** While many RL benchmarks feature partially observable environments, they tend to evaluate multiple agent capabilities simultaneously [2, 8, 27] (*e.g.*, exploration and modular skill learning), and may be solvable with a moderate memory capacity [14, 48]. Additionally, some benchmarks are designed for model-free agents [66, 37, 52], and may contain stochastic dynamics that are not controlled by the agents, making it hard to separately assess the memory capacity of world models. The recently proposed Memory Maze [50] focuses on measuring long-term memory and provides benchmark results for model-based agents. We build upon Memory Maze and introduce additional environments and tasks to probe a wider range of memory capabilities. Another recent work, TECO [67], also introduces datasets and a Transformer-based model for evaluating and improving long-term video prediction. Our work has a different focus than TECO in that we stress test models on extremely long sequences (up to 2000 steps), while TECO considers more visually complex environments, with sequence lengths capped at 300. Our experiment setup allows using relatively lightweight models to tackle significant challenges involving long-term memory. We include a comparison with TECO in Appendix F.

## 3 Background

S4 [21] and its variants [20, 40, 57, 15] are specialized parameterizations of linear state space models. We first present relevant background on linear state space models, and then introduce the S4 model.

**Linear State Space Models (SSMs)** are a widely used sequence model that defines a mapping from a 1-D input signal $u(t)$ to a 1-D output signal $y(t)$. They can be discretized into a sequence-to-sequence mapping by a step size $\Delta$. The continuous-time and discrete-time SSMs can be described as:

$$\text{(continuous-time)} \quad \begin{aligned} \boldsymbol{s}'(t) &= \boldsymbol{A}\boldsymbol{s}(t) + \boldsymbol{B}u(t) \\ y(t) &= \boldsymbol{C}\boldsymbol{s}(t) + \boldsymbol{D}u(t) \end{aligned}, \quad \text{(discrete-time)} \quad \begin{aligned} \boldsymbol{s}_k &= \overline{\boldsymbol{A}}\boldsymbol{s}_{k-1} + \overline{\boldsymbol{B}}u_k \\ y_k &= \overline{\boldsymbol{C}}\boldsymbol{s}_k + \overline{\boldsymbol{D}}u_k \end{aligned}. \quad (1)$$

Here, the vectors $\boldsymbol{s}(t)$ and $\boldsymbol{s}_k$ are the internal hidden state of the SSM, and the discrete-time matrices $\overline{\boldsymbol{A}}, \overline{\boldsymbol{B}}, \overline{\boldsymbol{C}}, \overline{\boldsymbol{D}}$ can be computed from their continuous-time counterparts $\boldsymbol{A}, \boldsymbol{B}, \boldsymbol{C}, \boldsymbol{D}$ and the step size $\Delta$. We will primarily deal with the discrete-time SSMs, which allow efficient autoregressive generation like RNNs due to the recurrence in $\boldsymbol{s}_k$.

Unlike RNNs, however, linear SSMs can offer parallelizable computation like Transformers. That is, given the input sequence $u_{1:T}$, the output sequence $y_{1:T}$ can be computed in parallel across time steps by a discrete convolution [21] or a parallel associative scan [57, 4]. In this work, we define the class of **Parallelizable SSMs (PSSMs)** to be the SSMs that provide the following interface for both parallel and single-step computation:

$$\text{(parallel)} \ \boldsymbol{y}_{1:T}, \boldsymbol{s}_T = \text{PSSM}(\boldsymbol{u}_{1:T}, \boldsymbol{s}_0), \quad \text{(single step)} \ \boldsymbol{y}_k, \boldsymbol{s}_k = \text{PSSM}(\boldsymbol{u}_k, \boldsymbol{s}_{k-1}), \quad (2)$$

where the inputs $\boldsymbol{u}_k$ and outputs $\boldsymbol{y}_k$ can be vectors.

**The S4 model** aims to use SSMs for deep sequence modeling, where the matrices $\boldsymbol{A}, \boldsymbol{B}, \boldsymbol{C}, \boldsymbol{D}$ and the step size $\Delta$ are learnable parameters to be optimized by gradient descent. Because SSMs involve computing powers of $\overline{\boldsymbol{A}}$, which is in general expensive and can lead to the exploding/vanishing gradients problem [49], SSMs with a randomly initialized $\boldsymbol{A}$ perform very poorly in practice [19].

To address these problems, S4 parameterizes $\boldsymbol{A}$ as a Diagonal Plus Low-Rank (DPLR) matrix [21, 17]: $\boldsymbol{A} = \boldsymbol{\Lambda} - \boldsymbol{P}\boldsymbol{P}^*$, where $\boldsymbol{\Lambda}$ is a diagonal matrix, $\boldsymbol{P}$ is typically a column vector (with rank 1), and $\boldsymbol{P}^*$ is the conjugate transpose of $\boldsymbol{P}$. This parameterization allows efficient computation of powers of $\overline{\boldsymbol{A}}$,

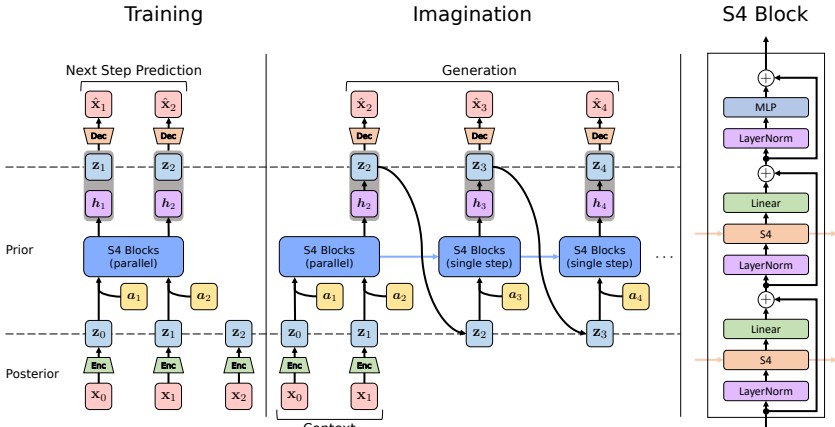

**Figure 1:** We propose S4WM, the first S4-based world model for improving long-term memory. S4WM efficiently models the long-range dependencies of environment dynamics in a compact latent space, using a stack of S4 blocks. This crucially allows fully parallelized training and fast recurrent latent imagination. S4WM is a general framework that is compatible with any parallelizable SSM including S5 and other S4 variants.

---

**Algorithm 1** S4WM Training

**Input:** $(\mathbf{x}_0, \boldsymbol{a}_1, \mathbf{x}_1, \ldots, \boldsymbol{a}_T, \mathbf{x}_T)$
    ▷ Obtain posterior latents
1: **for** time step $t = 0, \ldots, T$ **parallel do**
2:     $\mathbf{z}_t \sim q(\mathbf{z}_t \,|\, \mathbf{x}_t) = \text{Encoder}(\mathbf{x}_t)$
3: **end for**
    ▷ Prepare inputs to S4 blocks
4: **for** time step $t = 1, \ldots, T$ **parallel do**
5:     $\boldsymbol{g}_t = \text{MLP}(\text{concat}[\mathbf{z}_{t-1}, \boldsymbol{a}_t])$
6: **end for**
    ▷ Encode history by S4 blocks
7: $\boldsymbol{h}_{1:T}, \boldsymbol{s}_T = \text{S4Blocks}(\boldsymbol{g}_{1:T}, \boldsymbol{s}_0)$
    ▷ Compute prior and
      decode posterior latents
8: **for** time step $t = 1, \ldots, T$ **parallel do**
9:     $p(\mathbf{z}_t \,|\, \mathbf{z}_{<t}, \boldsymbol{a}_{\le t}) = \text{MLP}(\boldsymbol{h}_t)$
10:    $\hat{\mathbf{x}}_t = \text{Decoder}(\text{concat}[\boldsymbol{h}_t, \mathbf{z}_t])$
11: **end for**
12: Compute objective by Equation (9)

---

**Algorithm 2** S4WM Imagination

**Input:** context $(\mathbf{x}_{0:C}, \boldsymbol{a}_{1:C})$, query $\boldsymbol{a}_{C+1:T}$
    ▷ Encode context
1: **for** context step $t = 0, \ldots, C$ **parallel do**
2:     $\mathbf{z}_t \sim q(\mathbf{z}_t \,|\, \mathbf{x}_t) = \text{Encoder}(\mathbf{x}_t)$
3:     $\boldsymbol{g}_{t+1} = \text{MLP}(\text{concat}[\mathbf{z}_t, \boldsymbol{a}_{t+1}])$
4: **end for**
5: $\boldsymbol{h}_{1:C+1}, \boldsymbol{s}_{C+1} = \text{S4Blocks}(\boldsymbol{g}_{1:C+1}, \boldsymbol{s}_0)$
    ▷ Imagine in latent space
6: $\mathbf{z}_{C+1} \sim p(\mathbf{z}_{C+1} \,|\, \mathbf{z}_{0:C}, \boldsymbol{a}_{1:C+1}) = \text{MLP}(\boldsymbol{h}_{C+1})$
7: **for** query step $t = C + 2, \ldots, T$ **do**
8:     $\boldsymbol{g}_t = \text{MLP}(\text{concat}[\mathbf{z}_{t-1}, \boldsymbol{a}_t])$
9:     $\boldsymbol{h}_t, \boldsymbol{s}_t = \text{S4Blocks}(\boldsymbol{g}_t, \boldsymbol{s}_{t-1})$
10:    $\mathbf{z}_t \sim p(\mathbf{z}_t \,|\, \mathbf{z}_{<t}, \boldsymbol{a}_{\le t}) = \text{MLP}(\boldsymbol{h}_t)$
11: **end for**
    ▷ Decode imagined latents
12: **for** query step $t = C + 1, \ldots, T$ **parallel do**
13:    $\hat{\mathbf{x}}_t = \text{Decoder}(\text{concat}[\boldsymbol{h}_t, \mathbf{z}_t])$
14: **end for**

---

while also including the HiPPO matrices [18], which are theoretically derived based on continuous-time memorization and empirically shown to better capture long-range dependencies. In practice, S4 initializes $\boldsymbol{A}$ to the HiPPO matrix. To cope with vector-valued inputs and outputs (*i.e.*, $\boldsymbol{u}_k, \boldsymbol{y}_k \in \mathbb{R}^H$), S4 makes $H$ copies of the SSM, each operating on one dimension, and mixes the outputs by a position-wise linear layer. Follow-up works further simplify the parameterization of $\boldsymbol{A}$ to a diagonal matrix [20], and use a multi-input, multi-output SSM for vector-valued sequences [57].

## 4   S4WM: A General World Model for Parallelizable SSMs

We consider an agent interacting with a partially observable environment. At each time step $t$, the agent receives an image observation $\mathbf{x}_t$. It then chooses an action $\boldsymbol{a}_{t+1}$ based on its policy, and receives the next observation $\mathbf{x}_{t+1}$. For simplicity, we omit the reward here.

We aim to model $p(\mathbf{x}_{1:T} \,|\, \mathbf{x}_0, \boldsymbol{a}_{1:T})$, the distribution of future observations given the action sequence. We note that it is not required to model $p(\mathbf{x}_0)$, as world model imagination is typically conditioned

on at least one observation. While S4 and its variants have shown remarkable abilities to model long-range dependencies, they operate directly in the observation space. For example, S4 models images as sequences of pixels, and directly learns the dependencies among individual pixels. This is hard to scale to high-dimensional sequences, such as the sequences of images that we aim to model.

Inspired by RSSM [28], we propose S4WM, the first PSSM-based world model that learns the environment dynamics in a compact latent space. This not only allows fast parallelizable training, but also enables efficient modeling of long-range dependencies in the latent space. Importantly, S4WM is a general framework that can incorporate not only the specific S4 model [21] but also any PSSM defined by Equation (2), including S5 [57] and other variants [20, 40, 15]. It models the observations and state transitions through a probabilistic generative process:

$$p(\mathbf{x}_{1:T} \mid \mathbf{x}_0, \boldsymbol{a}_{1:T}) = \int p(\mathbf{z}_0 \mid \mathbf{x}_0) \prod_{t=1}^{T} p(\mathbf{x}_t \mid \mathbf{z}_{\leq t}, \boldsymbol{a}_{\leq t}) \, p(\mathbf{z}_t \mid \mathbf{z}_{<t}, \boldsymbol{a}_{\leq t}) \, \mathrm{d}\mathbf{z}_{0:T} \,, \tag{3}$$

where $\mathbf{z}_{0:T}$ are the stochastic latent states. We note that computing the likelihood $p(\mathbf{x}_t \mid \mathbf{z}_{\leq t}, \boldsymbol{a}_{\leq t})$ and the prior $p(\mathbf{z}_t \mid \mathbf{z}_{<t}, \boldsymbol{a}_{\leq t})$ requires extracting relevant information from the history $(\mathbf{z}_{<t}, \boldsymbol{a}_{\leq t})$. Therefore, it is crucial to maintain a long-term memory of the history. To this end, we use a stack of PSSM blocks to encode the history $(\mathbf{z}_{<t}, \boldsymbol{a}_{\leq t})$ into an embedding vector $\boldsymbol{h}_t$ for each $t$. This can be done in parallel during training and sequentially during imagination:

$$\textbf{(parallel)} \qquad \boldsymbol{h}_{1:T}, \boldsymbol{s}_T = \text{PSSM\_Blocks}(\boldsymbol{g}_{1:T}, \boldsymbol{s}_0) \,, \tag{4}$$

$$\textbf{(single step)} \qquad \boldsymbol{h}_t, \boldsymbol{s}_t = \text{PSSM\_Blocks}(\boldsymbol{g}_t, \boldsymbol{s}_{t-1}) \,. \tag{5}$$

Here, $\boldsymbol{g}_t = \text{MLP}(\texttt{concat}[\mathbf{z}_{t-1}, \boldsymbol{a}_t])$ is the input to the PSSM blocks, $\boldsymbol{s}_t$ is the collection of all internal hidden states, and $\boldsymbol{h}_t$ is the final output. In our experiments, we use S4 blocks (shown in Figure 1 Right), a particular instantiation of the PSSM blocks using the S4 model. We find that adding the final MLP in each S4 block can improve generation quality (see Appendix B.1 for an ablation and Figure 21 for a detailed illustration of S4 block architecture).

After obtaining $\boldsymbol{h}_t$, we use it to compute the sufficient statistics of the prior and likelihood:

$$p(\mathbf{z}_t \mid \mathbf{z}_{<t}, \boldsymbol{a}_{\leq t}) = \text{MLP}(\boldsymbol{h}_t) \,, \tag{6}$$

$$p(\mathbf{x}_t \mid \mathbf{z}_{\leq t}, \boldsymbol{a}_{\leq t}) = \mathcal{N}(\hat{\mathbf{x}}_t, \mathbf{1}) \,, \quad \hat{\mathbf{x}}_t = \text{Decoder}(\texttt{concat}[\boldsymbol{h}_t, \mathbf{z}_t]) \,. \tag{7}$$

For training, we use variational inference. The approximate posterior is defined as:

$$q(\mathbf{z}_{0:T} \mid \mathbf{x}_{0:T}, \boldsymbol{a}_{1:T}) = \prod_{t=0}^{T} q(\mathbf{z}_t \mid \mathbf{x}_t) \,, \quad \text{where } q(\mathbf{z}_0 \mid \mathbf{x}_0) = p(\mathbf{z}_0 \mid \mathbf{x}_0) \,. \tag{8}$$

We use a CNN encoder to compute the sufficient statistics of the posterior from image observations. This allows all posterior samples $\mathbf{z}_{0:T}$ to be obtained in parallel, thereby fully leveraging the parallel computation ability offered by PSSMs during training. We also provide an alternative design of the posterior in Appendix B.1. It is conditioned on the full history, and can obtain better generation quality at the cost of more computation.

The training objective is to maximize the evidence lower bound (ELBO):

$$\log p(\mathbf{x}_{1:T} \mid \mathbf{x}_0, \boldsymbol{a}_{1:T}) \geq \mathbb{E}_q \left[ \sum_{t=1}^{T} \log p(\mathbf{x}_t \mid \mathbf{z}_{\leq t}, \boldsymbol{a}_{\leq t}) - \mathcal{L}_{\text{KL}} \Big( q(\mathbf{z}_t \mid \mathbf{x}_t), \, p(\mathbf{z}_t \mid \mathbf{z}_{<t}, \boldsymbol{a}_{\leq t}) \Big) \right] \,. \tag{9}$$

In our experiments, we instantiate S4WM using the S4 model [21]. Figure 1 provides an illustration of training and imagination procedures, and Algorithms 1 and 2 provide detailed descriptions. Hyperparameters and further implementation details can be found in Appendix J.

## 5 Experiments

### 5.1 Environments

Unlike previous works [14, 48, 37, 52] that primarily evaluate the final performance of model-free agents on memory-demanding tasks, we seek to understand the memory capabilities of world models

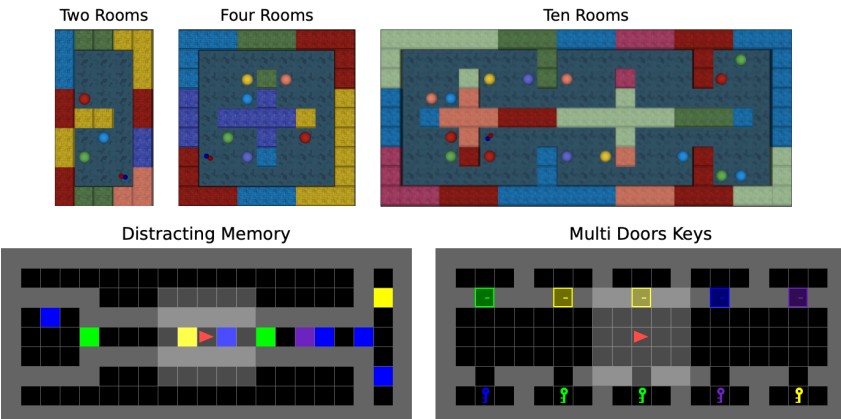

**Figure 2:** Partially observable 3D (Top) and 2D (Bottom) environments for evaluating memory capabilities of world models, including long-term imagination, context-dependent recall, reward prediction, and memory-based reasoning.

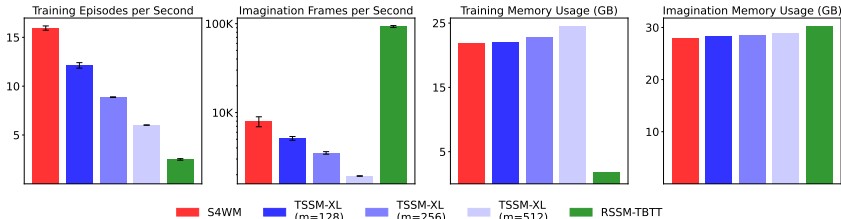

**Figure 3:** Comparison of speed and memory usage during training and imagination. S4WM is the fastest to train, while RSSM-TBTT is the most memory-efficient during training and has the highest throughput during imagination.

in model-based agents in terms of long-term imagination, context-dependent recall, reward prediction, and memory-based reasoning. We believe that our investigation provides more insights than the final performance alone, and paves the way for model-based agents with improved memory.

To this end, we develop a diverse set of environments shown in Figure 2, each targeting a specific memory capability. The environments are based on the 3D Memory Maze [50] and the 2D Mini-Grid [6], both with partial observations. The world models are learned from an offline dataset collected by a scripted policy for each environment. This allows the world models to be evaluated independently of the policy learning algorithms.

Specifically, for each episode, the environment is regenerated. To simplify the evaluation of world model imagination, we design the data collecting policy to consist of a *context* phase and a *query* phase. In the context phase, the policy fully traverses the environment, while in the query phase, the policy revisits some parts of the environment. For evaluation, we use unseen episodes collected by the same policy as training. The world model observes the context phase, and is then evaluated on its imagination given the action sequence in the query phase. Because the environments are deterministic and have moderate visual complexity, and the context phase fully reveals the information of the environment, it suffices to use the mean squared error (MSE) as our main evaluation metric.

In the following, we first motivate our choice of the baselines through a comparison of speed and memory consumption, and then introduce and present the results for each environment in detail.

## 5.2 Baselines

**RSSM-TBTT.** RSSM [28] is an RNN-based world model backbone used in state-of-the-art MBRL agents [29–31]. Recently, [50] show that training RSSM with truncated backpropagation through time (TBTT) can lead to better long-term memory ability. We follow their implementation and denote the model as RSSM-TBTT.

**Table 1:** Evaluation of long-term imagination. Each environment is labeled with (context steps | query steps). All models obtain good reconstruction, while S4WM is much better at long-term generation up to 500 steps. All models struggle in the Ten Rooms environment.

| | Two Rooms (301 \| 200) | | Four Rooms (501 \| 500) | | Ten Rooms (1101 \| 900) | |
| | Recon. MSE ($\downarrow$) | Gen. MSE ($\downarrow$) | Recon. MSE ($\downarrow$) | Gen. MSE ($\downarrow$) | Recon. MSE ($\downarrow$) | Gen. MSE ($\downarrow$) |
|---|---|---|---|---|---|---|
| RSSM-TBTT | **1.7** | 62.2 | **1.5** | 219.4 | **1.5** | 323.1 |
| TSSM-XL | 2.5 | 62.9 | 2.4 | 224.4 | 2.6 | 360.4 |
| S4WM | 1.8 | **27.3** | 1.7 | **44.0** | 1.8 | **224.4** |

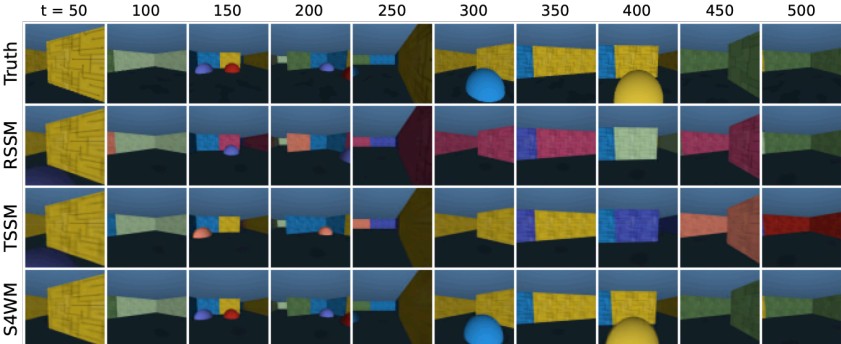

**Figure 4:** Long-term imagination in the Four Rooms environment. While RSSM-TBTT and TSSM-XL make many mistakes in wall colors, object colors, and object positions, S4WM is able to generate much more accurately, with only minor errors in object positions.

**TSSM-XL.** TSSM [5] is the first Transformer-based world model for improving long-term memory. It was originally evaluated on sequences of length $\sim 100$. In this paper, we seek to evaluate on much longer sequences (up to 2000 steps), and it is impractical to feed the entire sequence to the vanilla Transformer [61]. Therefore, we use Transformer-XL [9] as the backbone and denote the model as TSSM-XL. It divides the full sequence into chunks, and maintains a cache of the intermediate hidden states from processed chunks. This cache serves as an extended context that allows modeling longer-term dependencies.

**Speed and Memory Usage.** We note that the cache length $m$ is a crucial hyperparameter of TSSM-XL. A larger $m$ can potentially improve the memory capacity, at the cost of slower training and higher memory consumption. To ensure a fair comparison in our experiments in the next few sections, we first investigate the speed and memory usage of S4WM, RSSM-TBTT, and TSSM-XL with several $m$ values. Details can be found in Appendix I. As shown in Figure 3, S4WM and TSSM-XL trains much faster than RSSM-TBTT due to their parallel computation during training, while RSSM-TBTT is much more memory-efficient. For imagination, RSSM-TBTT achieves $\sim 10\times$ throughput compared to S4WM and TSSM-XL. While S4WM also uses recurrence during imagination, its multi-layered recurrence structure with MLPs in between slows down its performance. As for memory usage, the decoder takes up most of the memory decoding all steps in parallel, leading to similar memory consumption of all models.

Based on our investigation, TSSM-XL with a cache length $m = 128$ is the closest to S4WM in terms of speed and memory usage. Therefore, we use TSSM-XL with $m = 128$ for all subsequent experiments. We provide a more thorough investigation with larger cache lengths in Appendix B.2.

## 5.3 Long-Term Imagination

The ability of world models to perform long-term imagination is crucial to long-horizon planning. While many RL benchmarks can be tackled with short-term imagination of $\sim 15$ steps [31], here we seek to understand the long-term imagination capability of world models and explore their limits by letting the world models imagine hundreds of steps into the future.

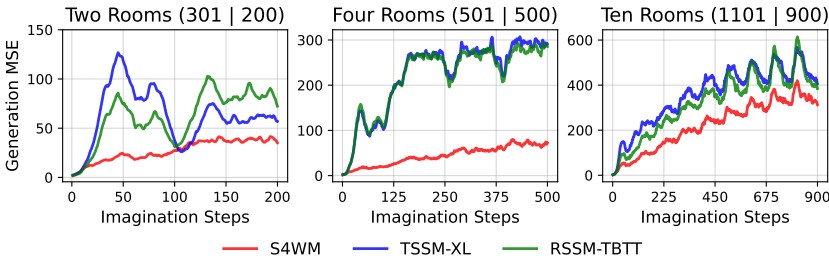

**Figure 5:** Generation MSE per imagination step. Each environment is labeled with (context steps | query steps). S4WM maintains a relatively good generation quality for up to 500 steps, while RSSM-TBTT and TSSM-XL make large generation errors even within 50 steps.

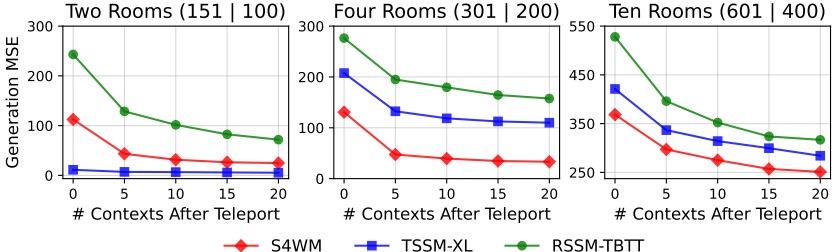

**Figure 6:** Evaluation of context-dependent recall in teleport environments. Each environment is labeled with (context steps | query steps). We provide up to 20 observations after the teleport as additional contexts. TSSM-XL performs the best in the Two Rooms environment where the context phase is short, and is able to recall successfully without additional observations. When the context phase is longer, S4WM performs the best.

To this end, we develop three environments with increasing difficulty, namely Two Rooms, Four Rooms, and Ten Rooms, based on the 3D Memory Maze [50]. The top-down views are shown in Figure 2. In the context phase, the data collecting policy starts from a random room, sequentially traverses all rooms, and returns to the starting room. In the query phase, the policy revisits each room in the same order as the context phase.

As shown in Table 1, all models obtain good reconstruction, while S4WM is much better in the Two Rooms and Four Rooms environment for long-term generation up to 500 steps. We demonstrate the superior generation quality of S4WM in Figure 4. All models are able to capture the high-level maze layout. However, RSSM-TBTT and TSSM-XL make many mistakes in details such as wall colors, object colors, and object positions, while S4WM is able to generate much more accurately, with only minor errors in object positions. We further show the per step generation MSE in Figure 5. S4WM is able to maintain a relatively good generation quality for up to 500 steps, while RSSM-TBTT and TSSM-XL make large generation errors even within 50 steps. We notice a periodic drop in the generation MSE. This is when the agent moves from one room to another through a narrow corridor where the action sequence is less diverse.

We also find that all models struggle in the Ten Rooms environment where the context length is 1101 and the query length is 900. This likely reaches the sequence modeling limits of the S4 model, and we leave the investigation of more sophisticated model architectures to future work.

### 5.4 Context-Dependent Recall

Humans are able to recall past events in great detail. This has been compared to "mental time travel" [60, 58, 37]. Motivated by this, we develop a "teleport" version of the Two Rooms, Four Rooms, and Ten Rooms environments. After the initial context phase, the agent is teleported to a random point in history, and is asked to recall what happened from that point onwards, given the exact same action sequence that the agent took.

To succeed in this task, the agent needs to figure out where it is teleported by comparing the new observations received after the teleport to its own memory of the past. In other words, the content

**Table 2:** Reward prediction accuracy in the Distracting Memory environments. Each environment is labeled with (context steps | query steps). S4WM succeeds in all environments. TSSM-XL has limited success when observing the full sequence, but fails to predict rewards within imagination. RSSM-TBTT completely fails.

| | Width = 100 (199 \| 51) | | Width = 200 (399 \| 101) | | Width = 400 (799 \| 201) | |
|---|---|---|---|---|---|---|
| | Inference Accuracy (↑) | Imagination Accuracy (↑) | Inference Accuracy (↑) | Imagination Accuracy (↑) | Inference Accuracy (↑) | Imagination Accuracy (↑) |
| RSSM-TBTT | 47.9% | 49.6% | 48.7% | 48.4% | 50.4% | 52.2% |
| TSSM-XL | **100.0%** | 51.2% | 99.9% | 51.3% | 50.4% | 51.0% |
| S4WM | **100.0%** | **100.0%** | **100.0%** | **100.0%** | **100.0%** | **100.0%** |

**Figure 7:** Imagination in the Distracting Memory environment of width 100. RSSM-TBTT and TSSM-XL fail to keep track of the agent's position, leading to inaccurate reward prediction within imagination.

to recall depends on the new observations. Transformers have been shown to be better than S4 at performing such context-dependent operations in low-dimensional sequence manipulation tasks [24] and synthetic language modeling tasks [15]. We investigate this in the context of world model learning, with high-dimensional image inputs.

To help the model retrieve the correct events in history, we provide up to 20 observations after the teleport as additional contexts. The generation MSE of the recall is reported in Figure 6. TSSM-XL performs the best in the Two Rooms environment where the context phase is short, and is able to recall successfully without additional observations. When the context phase is longer as in Four Rooms and Ten Rooms, S4WM performs the best. We visually show the recall quality with 20 observations after the teleport in Figure 18. In the Two Rooms environment, both TSSM-XL and S4WM are able to recall accurately. However, only S4WM is able to maintain such recall quality in the more challenging Four Rooms environment.

## 5.5 Reward Prediction

To facilitate policy learning within imagination, world models need to accurately predict the rewards. In this section, we evaluate the reward prediction accuracy over long time horizons. To decouple the challenges posed by 3D environments from long-term reward prediction, we use the visually simpler 2D MiniGrid [6] environment.

Specifically, we develop the Distracting Memory environment, which is more challenging than the original MiniGird Memory environment, due to distractors of random colors being placed in the hallway. A top-down view is shown in Figure 2. Each episode terminates when the agent reaches one of the squares on the right. A reward of 1 is given if the square reached is of the same color as the square in the room on the left. Otherwise, no reward is given. In the context phase, the data collecting policy starts in the middle of the hallway, then traverses the hallway and returns to the starting position. In the query phase, the policy goes to one of the two squares on the right uniformly at random. To accurately predict the reward, the world model must learn to ignore the distractors while keeping track of the agent's position.

We report two types of reward prediction accuracy in Table 2. The inference accuracy is measured when the model takes the full sequence of observations as input (including both the context and the query phases). This evaluates the model's ability to capture long-range dependencies independently of the imagination quality. In contrast, the imagination accuracy is evaluated within the model's imagination, conditioned on the observations in the context phase and additionally the action sequence in the query phase.

**Table 3:** Memory-based reasoning in the Multi Doors Keys environments. Each environment is labeled with (context steps | query steps). S4WM performs well on all environments, while others struggle.

| | Three Keys (76 \| 174) | | Five Keys (170 \| 380) | | Seven Keys (296 \| 654) | |
|---|---|---|---|---|---|---|
| | Recon. MSE ($\downarrow$) | Gen. MSE ($\downarrow$) | Recon. MSE ($\downarrow$) | Gen. MSE ($\downarrow$) | Recon. MSE ($\downarrow$) | Gen. MSE ($\downarrow$) |
| RSSM-TBTT | 0.05 | 5.16 | 0.04 | 6.36 | 0.03 | 6.28 |
| TSSM-XL | 0.05 | 1.27 | 0.03 | 6.05 | **0.02** | 9.24 |
| S4WM | **0.01** | **0.04** | **0.02** | **0.27** | **0.02** | **0.10** |

Our results show that only S4WM is able to accurately predict rewards within imagination. TSSM-XL has limited success when observing the full sequence, but fails to imagine future rewards accurately. RSSM-TBTT completely fails, and its reward prediction is close to random guessing. Our visualization of model imagination in Figure 7 reveals that the failure of TSSM-XL and RSSM-TBTT is mainly due to their inability to keep track of the agent's position.

## 5.6 Memory-Based Reasoning

In the previous experiments, the model's memory of the environment can largely be kept fixed after the context phase. In this section, we explore the setting where the memory needs to be frequently updated in order to reason about the future.

We develop the Multi Doors Keys environment, where the agent collects keys to unlock doors. A top-down view is shown in Figure 2. Each time a door is unlocked, the corresponding key will be consumed, so it cannot be used to unlock other doors of the same color. The agent is allowed to possess multiple keys. In the context phase, the agent visits all keys and doors without picking up any keys. In the query phase, the agent attempts to unlock two random doors after picking up each key. After all keys are picked up, the agent will try to unlock each door once again. To successfully predict the outcome when the agent attempts to unlock a door, the world model must constantly update its memory when a key is picked up or consumed.

Since the environment is visually simple, we find the generation MSE to be a good indicator of how well the model predicts the future door states. As reported in Table 3 and visually shown in Figure 19, S4WM performs well on all environments, demonstrating its ability to keep updating the memory, while both RSSM-TBTT and TSSM-XL struggle.

## 6 Conclusion

In this paper, we introduced S4WM, the first PSSM-based visual world model that effectively expands the long-range sequence modeling ability of S4 and its variants from low-dimensional inputs to high-dimensional images. Furthermore, we presented the first comparative investigation of major world model backbones in a diverse set of environments specifically designed to evaluate critical memory capabilities. Our findings demonstrate the superior performance of S4WM over RNNs and Transformers across multiple tasks, including long-term imagination, context-dependent recall, reward prediction, and memory-based reasoning.

**Limitations and Future Work.** We primarily focused on visually simple and deterministic environments to limit the computation cost and simplify the evaluation process. Future work could explore more sophisticated model architectures and proper evaluation metrics for complex and stochastic environments. In addition, we mainly evaluated imagination quality and did not test world models in conjunction with policy learning. Future work could develop and thoroughly test MBRL agents based on S4WM. To demonstrate the potential of S4WM for policy learning, we provide offline probing results in Appendix D and conduct a skill-level MPC experiment in Appendix E. We find that S4WM outperforms RSSM in offline probing when it is instantiated with S5, and leads to higher task success rates when used for planning. In Appendix G, we additionally demonstrate that the long-term imagination quality can be further improved by instantiating S4WM with S5, showing the potential of our general S4WM framework for incorporating more advanced parallelizable SSMs.

## Acknowledgments and Disclosure of Funding

This work is supported by Brain Pool Plus Program (No. 2021H1D3A2A03103645) through the National Research Foundation of Korea (NRF) funded by the Ministry of Science and ICT. We thank Jurgis Pašukonis, Danijar Hafner, Chang Chen, Jaesik Yoon, and Caglar Gulcehre for insightful discussions.

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

# A  Environment and Dataset Details

For each 3D environment (*i.e.*, Two Rooms, Four Rooms, and Ten Rooms), we generate $30\text{K}$ trajectories using a scripted policy, of which $28\text{K}$ are used for training, $1\text{K}$ for validation, and $1\text{K}$ for testing. For each 2D environment (*i.e.*, Distracting Memory and Multi Doors Keys), we generate $10\text{K}$ trajectories using a scripted policy, of which $8\text{K}$ are used for training, $1\text{K}$ for validation, and $1\text{K}$ for testing. All reported results are obtained from the test trajectories, using the model checkpoints that achieve the best validation loss. The image observations are of size $64\times64\times3$ for 3D environments, and $40\times40\times3$ for 2D environments.

# B  Ablation Study

In this section, we investigate alternative architectural choices for S4WM and different cache lengths $m$ for TSSM-XL. We conduct these ablation studies on the Four Rooms and Ten Rooms environments.

## B.1  Alternative Architectures of S4WM

**S4WM-Full-Posterior.** In our main experiments, we have chosen to use the factorized posterior

$$q(\mathbf{z}_{0:T} \mid \mathbf{x}_{0:T}, \boldsymbol{a}_{1:T}) = \prod_{t=0}^{T} q(\mathbf{z}_t \mid \mathbf{x}_t) \tag{10}$$

for simplicity and parallel training ability. However, we note that it is possible to condition on the full history while maintaining the parallel training ability:

$$q(\mathbf{z}_{0:T} \mid \mathbf{x}_{0:T}, \boldsymbol{a}_{1:T}) = \prod_{t=0}^{T} q(\mathbf{z}_t \mid \mathbf{x}_{\leq t}, \boldsymbol{a}_{\leq t}) . \tag{11}$$

We illustrate this architecture in Figure 8. Here, we first use a CNN encoder to obtain a deterministic embedding $\boldsymbol{e}_t$ for each image observation $\mathbf{x}_t$, and then use a stack of S4 blocks to encode the history and compute the sufficient statistics of the posterior for all $t$ in parallel:

$$q(\mathbf{z}_t \mid \mathbf{x}_{\leq t}, \boldsymbol{a}_{\leq t}) = \text{MLP}(\boldsymbol{h}_t) , \quad \boldsymbol{h}_{0:T}, \boldsymbol{s}_T = \text{S4Blocks}(\boldsymbol{g}_{0:T}, \boldsymbol{s}_{-1}) , \quad \boldsymbol{g}_t = \text{MLP}(\texttt{concat}[\boldsymbol{e}_t, \boldsymbol{a}_t]) . \tag{12}$$

We have defined the dummy action $\boldsymbol{a}_0 = \varnothing$ and the initial S4 hidden state $\boldsymbol{s}_{-1}$, which are both implemented as vectors of all zeros. We note that the S4 blocks in the posterior are not shared with those in the prior.

**S4WM-No-MLP.** In our implementation, each S4 block consists of two S4 layers and one MLP. We note that the MLP is not used in the original S4 [21] for the Long Range Arena tasks [59], but is commonly used in language modeling and audio generation [17]. Hence, we consider a model variant without the MLP in the S4 blocks to investigate the importance of this MLP in world model learning.

**Results.** We report results on the non-teleport Four Rooms and Ten Rooms environments in Table 4 and Figures 9 and 10. Results on teleport environments are reported in Figures 11 and 12. We also show a comparison of speed and memory usage in Figure 13. Our results suggest that S4WM-Full-Posterior performs similarly as S4WM on the Four Rooms environment, and becomes better in the more challenging Ten Rooms environment where the episode length is longer. However, it is more computationally demanding than S4WM during training. In contrast, while S4WM-No-MLP is the most computationally efficient, its performance is much worse than S4WM, indicating the importance of the MLP in S4 blocks to both long-term imagination and context-dependent recall.

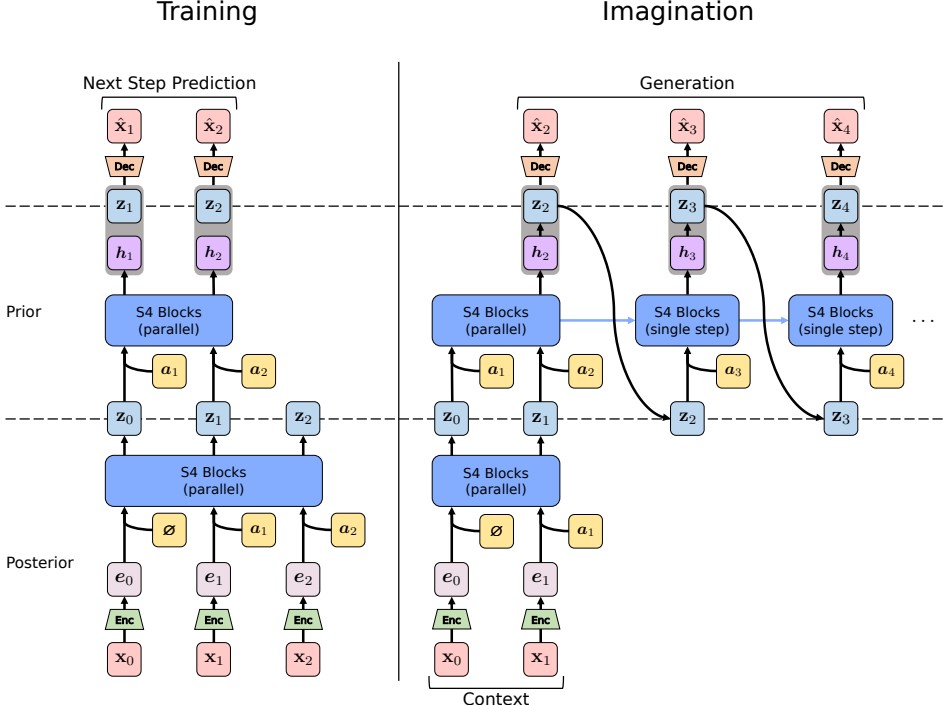

**Figure 8:** Architecture of S4WM-Full-Posterior. It maintains the parallel training ability while computing the posterior from the full history. ∅ denotes dummy actions.

**Table 4:** Comparison of alternative S4WM architectures on long-term imagination. Each environment is labeled with (context steps | query steps). S4WM-Full-Posterior is comparable to S4WM on the Four Rooms environment, but is better on the more challenging Ten Rooms environment. In contrast, S4WM-No-MLP performs much worse, suggesting the importance of the MLP in the S4 blocks to long-term imagination.

| | Four Rooms (501 \| 500) | | Ten Rooms (1101 \| 900) | |
| --- | --- | --- | --- | --- |
| | Recon. MSE ($\downarrow$) | Gen. MSE ($\downarrow$) | Recon. MSE ($\downarrow$) | Gen. MSE ($\downarrow$) |
| S4WM | **1.7** | 44.0 | **1.8** | 224.4 |
| S4WM-Full-Posterior | **1.7** | **38.0** | 2.0 | **171.1** |
| S4WM-No-MLP | 2.3 | 68.6 | 2.5 | 277.0 |

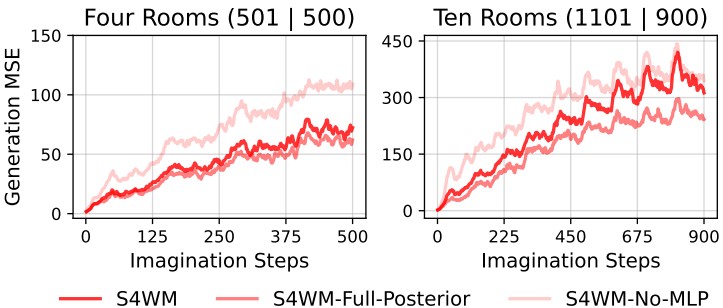

**Figure 9:** Generation MSE per imagination step of alternative S4WM architectures. Each environment is labeled with (context steps | query steps). S4WM-Full-Posterior performs the best as imagination horizon increases.

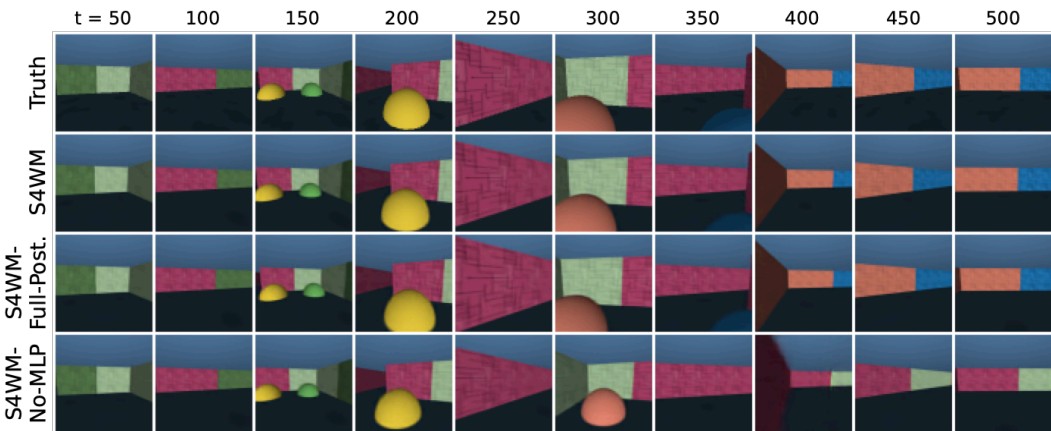

**Figure 10:** Long-term imagination from alternative S4WM architectures in the Four Rooms environment. S4WM and S4WM-Full-Posterior have similar imagination quality in this environment, while S4WM-No-MLP makes more mistakes in wall colors and object positions.

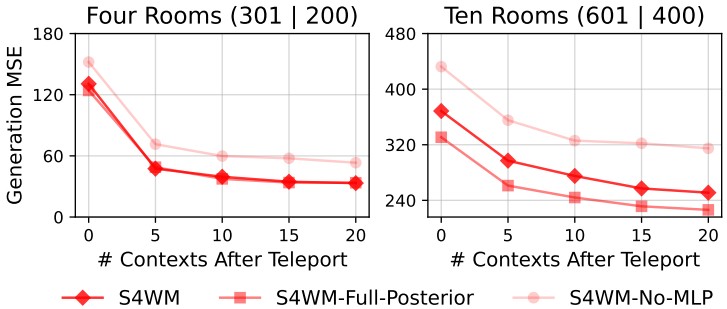

**Figure 11:** Comparison of alternative S4WM architectures on context-dependent recall in teleport environments. Each environment is labeled with (context steps | query steps). We provide up to 20 observations after the teleport as additional contexts. S4WM-Full-Posterior performs similarly as S4WM when the context phase is short, and becomes better when the context phase is longer. S4WM-No-MLP performs the worst, suggesting the importance of the MLP in the S4 blocks to context-dependent recall.

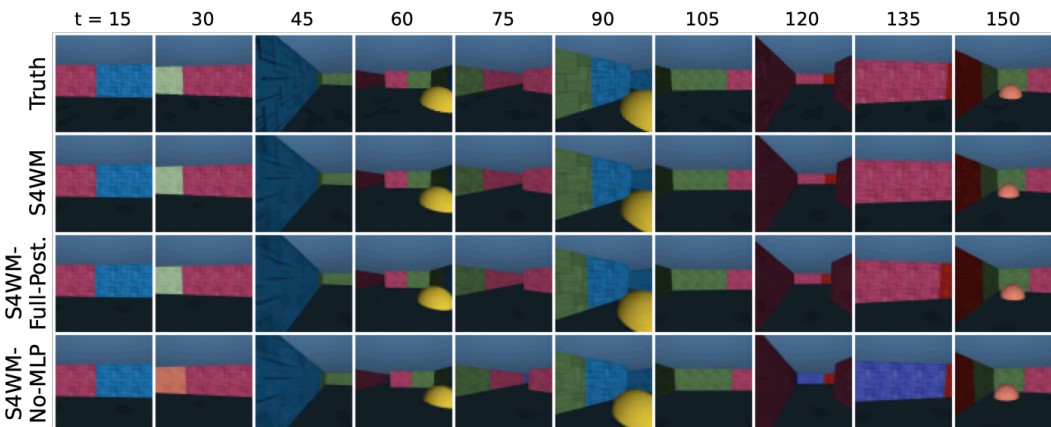

**Figure 12:** Context-dependent recall from alternative S4WM architectures in the Four Rooms teleport environment. 20 observations after the teleport are provided as additional contexts. S4WM and S4WM-Full-Posterior perform similarly in this environment, while S4WM-No-MLP makes more mistakes in wall colors and object positions.

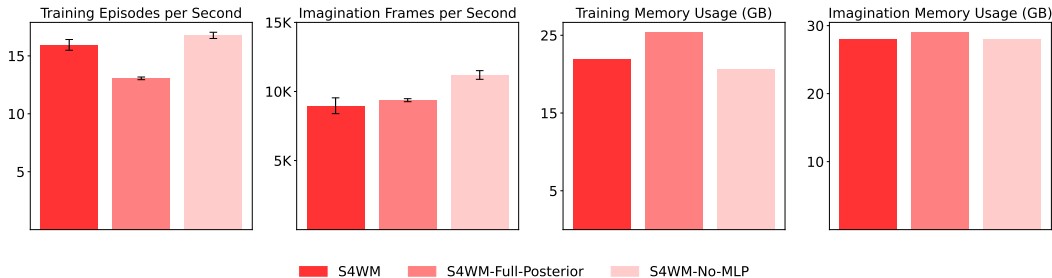

**Figure 13:** Comparison of speed and memory usage for alternative S4WM architectures. S4WM-Full-Posterior is more computationally demanding than S4WM during training, while S4WM-No-MLP is the most efficient.

**Table 5:** Comparison of TSSM-XL with different cache lengths $m$ on long-term imagination. Each environment is labeled with (context steps | query steps). Larger cache lengths show better generation quality, at the cost of more computation.

| | Four Rooms (501 | 500) | | Ten Rooms (1101 | 900) | |
|---|---|---|---|---|
| | Recon. MSE ($\downarrow$) | Gen. MSE ($\downarrow$) | Recon. MSE ($\downarrow$) | Gen. MSE ($\downarrow$) |
| TSSM-XL ($m = 128$) | **2.4** | 224.4 | 2.6 | 360.4 |
| TSSM-XL ($m = 256$) | **2.4** | 124.4 | **2.5** | 295.1 |
| TSSM-XL ($m = 512$) | **2.4** | **38.4** | **2.5** | **277.6** |

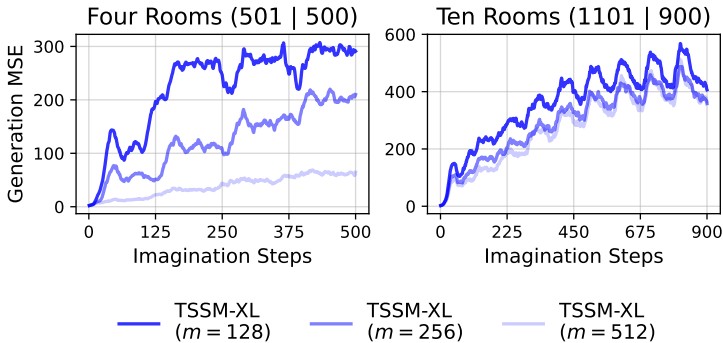

**Figure 14:** Generation MSE per imagination step of TSSM-XL with different cache lengths $m$. Each environment is labeled with (context steps | query steps). Increasing the cache length helps improving long-term imagination.

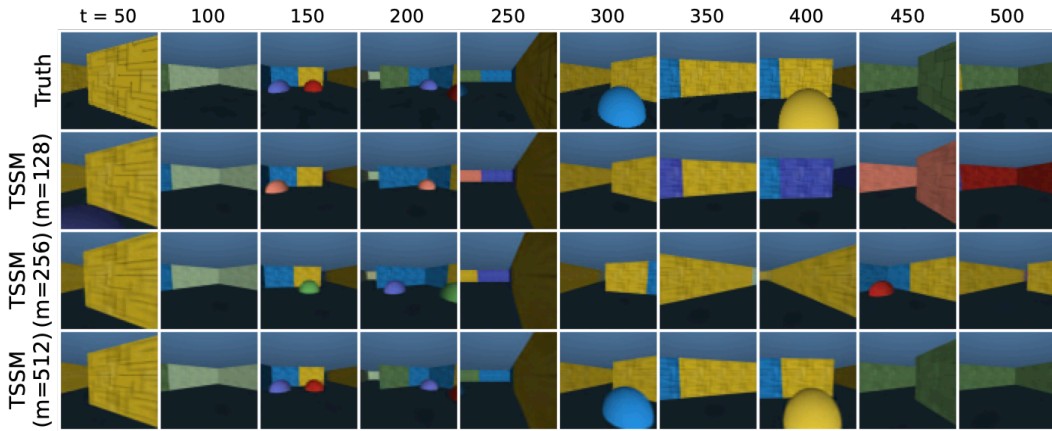

**Figure 15:** Long-term imagination from TSSM-XL with different cache lengths $m$ in the Four Rooms environment. TSSM-XL ($m = 512$) works well in this environment and is comparable to S4WM, while TSSM-XL ($m = 128$) and TSSM-XL ($m = 256$) make many mistakes.

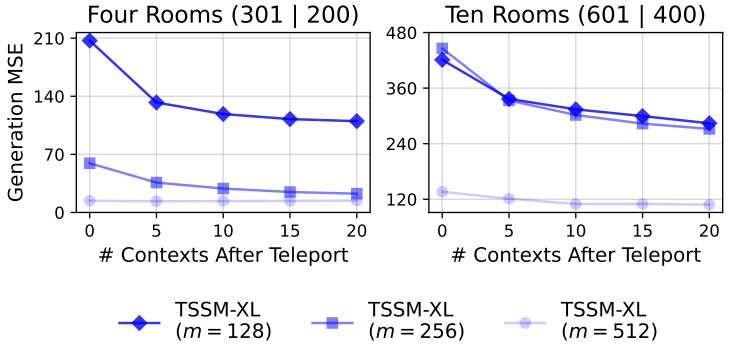

**Figure 16:** Comparison of TSSM-XL with different cache lengths $m$ on context-dependent recall in teleport environments. Each environment is labeled with (context steps | query steps). We provide up to 20 observations after the teleport as additional contexts. Increasing the cache length improves context-dependent recall. With a sufficiently large cache length, TSSM-XL without observing additional contexts can outperform S4WM.

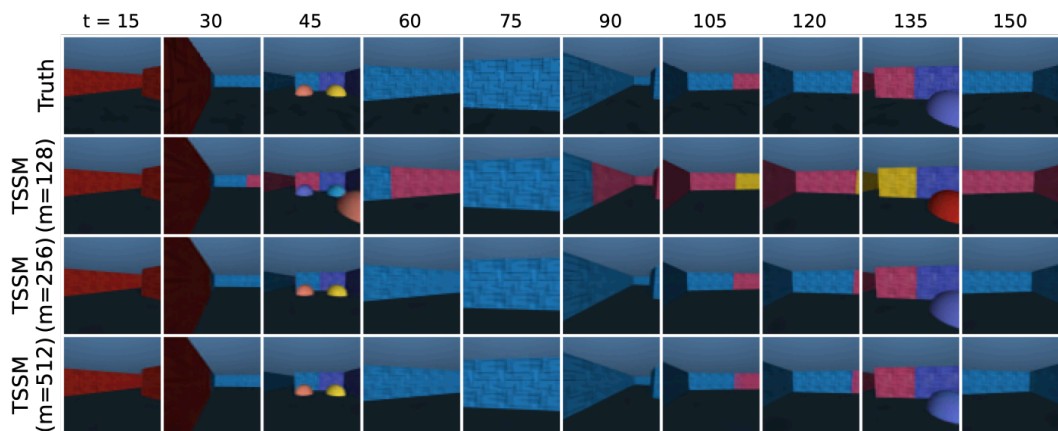

**Figure 17:** Context-dependent recall from TSSM-XL with different cache lengths $m$ in the Four Rooms teleport environment. 20 observations after the teleport are provided as additional contexts. TSSM-XL ($m = 256$) and TSSM-XL ($m = 512$) perform similarly in this environment, and are both better than TSSM-XL ($m = 128$).

## B.2 Cache Length of TSSM-XL

In our main experiments, we have used the cache length $m = 128$ for TSSM-XL, because it is close to S4WM in terms of computational cost. Here we provide a more thorough investigation with larger cache lengths.

We report results on the non-teleport Four Rooms and Ten Rooms environments in Table 5 and Figures 14 and 15. Results on teleport environments are reported in Figures 16 and 17. We find that increasing the cache length generally improves generation quality, at the cost of slower training and imagination speed. Notably, TSSM-XL with $m = 512$ shows better context-dependent recall than S4WM on the Ten Rooms teleport environment, consistent with the findings in previous work [24, 15] that Transformers are better than S4 at performing context-dependent operations.

# C  Additional Experiment Figures

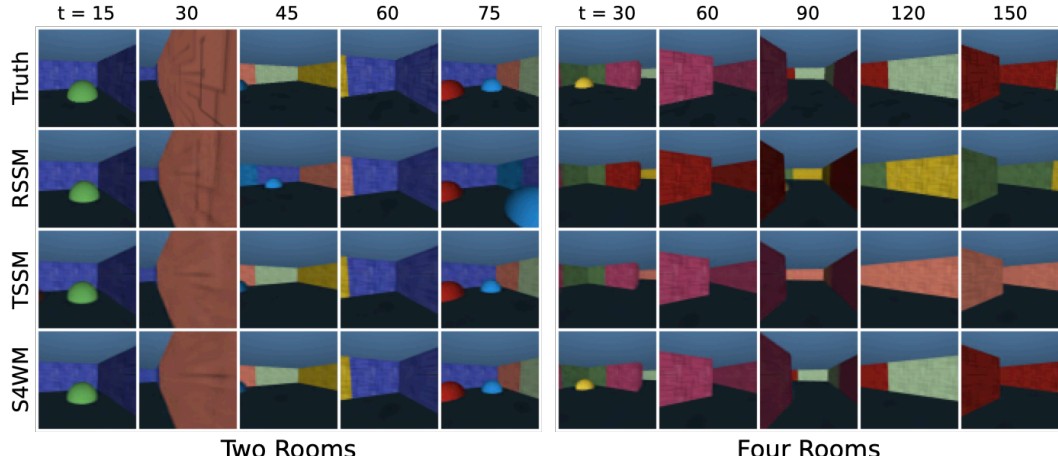

**Figure 18:** Context-dependent recall in the teleport environments. 20 observations after the teleport are provided as additional contexts. TSSM-XL and S4WM perform similarly in the Two Rooms environment, while only S4WM is able to maintain good recall quality in the Four Rooms environment.

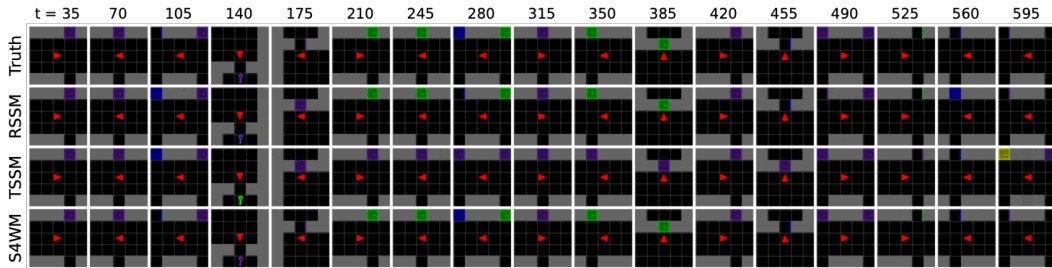

**Figure 19:** Imagination in the Multi Doors Keys environment with seven keys.

# D  Offline Probing on Memory Maze

Pašukonis *et al.* [50] recently proposed the Memory Maze offline probing benchmark for evaluating the representation learning ability of world models. For completeness, we report the benchmark results in Table 6. Our implementation is based on the newer DreamerV3 [31], and the results of RSSM-TBTT are slightly better than reported in the original Memory Maze paper.

For RSSM-TBTT, TSSM-XL, and S4WM, we use $\texttt{concat}[\boldsymbol{h}_t, \mathbf{z}_t]$ as the input to the probing network, where $\boldsymbol{h}_t$ is the output of the final Transformer/S4 block for TSSM-XL and S4WM. The probing network is an MLP with 4 hidden layers, each consisting of 1024 hidden units and followed by layer normalization [1] and SiLU [53] nonlinearity.

**Table 6:** Memory Maze offline probing benchmark results.

|  | Memory 9×9 Walls (Acc. ↑) | Memory 15×15 Walls (Acc. ↑) | Memory 9×9 Objects (MSE ↓) | Memory 15×15 Objects (MSE ↓) |
|---|---|---|---|---|
| Constant Baseline | 80.8% | 78.3% | 23.9 | 64.8 |
| RSSM-TBTT | 95.0% | 81.7% | 5.4 | 32.6 |
| TSSM-XL | 86.4% | 79.8% | 10.2 | 35.1 |
| S4WM | 88.4% | 80.2% | 8.9 | 34.8 |
| S5WM | **98.3%** | **81.8%** | **1.8** | **25.3** |

Both TSSM-XL and S4WM underperform RSSM-TBTT. We conjecture that this is because in RSSM-TBTT, $h_t$ captures more global information, while in TSSM-XL and S4WM, $h_t$ captures more local information for each $t$.

Comparing RSSM-TBTT and S4WM, we find that the S4 hidden state $s_t$ plays a similar role as the $h_t$ in RSSM-TBTT by carrying over information from previous time steps. Therefore, we conjecture that $s_t$ will capture more global information and $\texttt{concat}[s_t, z_t]$ can be a more suitable embedding for probing. However, the S4 hidden state $s_t$ has a much higher (typically $64\times$) dimension than $h_t$, making the above solution impractical.

Fortunately, S4WM is a general framework that works with many S4 variants. In particular, S5 [57] uses a hidden state $s_t$ that is of similar dimension as $h_t$. Hence, we replace the S4 layers in S4WM with S5 layers, and call this model S5WM. We extract the S5 hidden state of the last S5 layer in each S5 block, and concatenate them with $z_t$ to form the input to the probing network. Our results show that this model outperforms RSSM-TBTT by a large margin on the $9\times9$ maze. Nevertheless, the larger $15\times15$ maze remains challenging for all models.

# E  Skill-Level Model-Predictive Control

To show the potential of S4WM for planning, we consider a skill-level Model-Predictive Control (MPC) agent, with predefined skills and an offline-trained world model. Specifically, we consider the Multi Doors Keys environments. The predefined skills are (1) picking up a key at a specific position, and (2) going to a door at a specific position and attempting to unlock it. The agent will return to its starting position at the end of each skill. The task is to unlock a door specified by a goal image, which shows the door in unlocked state (see Figure 20 for an illustration). The agent is allowed to execute two skills: pick up one key and then attempt to unlock one door. Hence, planning has two stages. First, the agent enumerates all allowed skill sequences (of length 2), and uses the world model to predict which skill sequence is most likely to unlock the correct door. Then the agent executes the first skill in the environment and replans. We show the success rate of RSSM-TBTT, TSSM-XL, and S4WM in Table 7. The MPC agent equipped with S4WM is able to unlock every door in all three environments, while RSSM-TBTT and TSSM-XL succeed roughly half of the time.

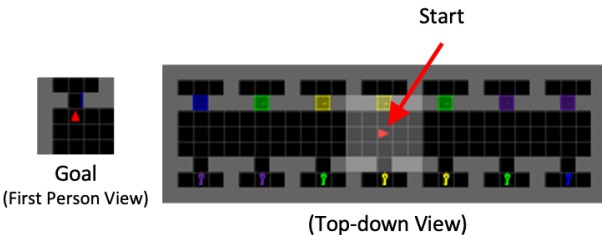

**Figure 20:** Illustration of the task solved by the skill-level MPC agent. The task is to unlock the door specified by the goal image on the left. The agent is allowed to execute two skills: pick up one key and then attempt to unlock one door. We use MPC to plan the optimal skill sequence with the offline-trained world model.

**Table 7:** Success rate of skill-level MPC agents in the Multi Doors Keys environments.

|  | Three Keys | Five Keys | Seven Keys |
|---|---|---|---|
| RSSM-TBTT | 0% | 60% | 42.86% |
| TSSM-XL | **100**% | 40% | 57.14% |
| S4WM | **100**% | **100**% | **100**% |

**Table 8:** Comparison of generation quality on DMLab following TECO evaluation protocol.

| | DMLab | | | |
|---|---|---|---|---|
| | PSNR ($\uparrow$) | SSIM ($\uparrow$) | LPIPS ($\downarrow$) | #Params |
| CW-VAE [55] | 12.6 | 0.372 | 0.465 | 111M |
| Latent FDM [32] | 17.8 | 0.588 | 0.222 | 31M |
| S4WM | 20.6 | 0.667 | 0.196 | 41M |
| TECO [67] | **21.9** | **0.703** | **0.157** | 169M |

**Table 9:** Evaluation of long-term imagination. Each environment is labeled with (context steps | query steps).

| | Two Rooms (301 | 200) | | Four Rooms (501 | 500) | | Ten Rooms (1101 | 900) | |
|---|---|---|---|---|---|---|
| | Recon. MSE ($\downarrow$) | Gen. MSE ($\downarrow$) | Recon. MSE ($\downarrow$) | Gen. MSE ($\downarrow$) | Recon. MSE ($\downarrow$) | Gen. MSE ($\downarrow$) |
| S4WM | 1.8 | 27.3 | 1.7 | 44.0 | 1.8 | 224.4 |
| S5WM | **1.5** | **10.6** | **1.3** | **19.2** | **1.3** | **159.2** |

## F   Comparison with TECO

We provide a quantitative comparison with TECO [67] and relevant baselines in Table 8. This comparison is done on DMLab following the TECO evaluation protocol. TECO and other baselines use a pretrained VQGAN [13] encoder/decoder, while S4WM uses a jointly trained simple CNN encoder/decoder (see Appendix J and Table 10 for architecture details and hyperparameters). S4WM significantly outperforms Clockwork VAE [55] and Latent FDM [32] (a diffusion-based model), and is close to TECO despite having much fewer parameters.

## G   Additional Results of Instantiating S4WM with S5

We show a comparison of long-term imagination quality between S4WM and S5WM in Table 9, where S5WM is our instantiation of S4WM with the S5 model. S5WM demonstrates even better imagination quality than S4WM, showing the potential of our general framework for incorporating more advanced parallelizable SSMs.

## H   Extended Background

In this section, we briefly introduce RSSM and TSSM for completeness. We denote the sequence of observations and actions as $(\mathbf{x}_0, \boldsymbol{a}_1, \mathbf{x}_1, \boldsymbol{a}_2, \mathbf{x}_2, \ldots, \boldsymbol{a}_T, \mathbf{x}_T)$. Namely, the agent takes action $\boldsymbol{a}_{t+1}$ after observing $\mathbf{x}_t$, and receives the next observation $\mathbf{x}_{t+1}$. We omit the reward for simplicity.

### H.1   RSSM

RSSM [28] models the observations and state transitions through the following generative process:

$$p(\mathbf{x}_{0:T} \mid \boldsymbol{a}_{1:T}) = \int \prod_{t=0}^{T} p(\mathbf{x}_t \mid \mathbf{z}_{\leq t}, \boldsymbol{a}_{\leq t}) \, p(\mathbf{z}_t \mid \mathbf{z}_{<t}, \boldsymbol{a}_{\leq t}) \, d\mathbf{z}_{0:T} \,, \tag{13}$$

where $\mathbf{z}_{0:T}$ are the stochastic latent states. The approximate posterior is defined as:

$$q(\mathbf{z}_{0:T} \mid \mathbf{x}_{0:T}, \boldsymbol{a}_{1:T}) = \prod_{t=0}^{T} q(\mathbf{z}_t \mid \mathbf{z}_{<t}, \boldsymbol{a}_{\leq t}, \mathbf{x}_t) \,. \tag{14}$$

The conditioning on previous states $\mathbf{z}_{<t}$ and actions $\boldsymbol{a}_{\leq t}$ appears multiple times. RSSM uses a shared GRU [7] to compress $\mathbf{z}_{<t}$ and $\boldsymbol{a}_{\leq t}$ into a deterministic encoding $\boldsymbol{h}_t$:

$$\boldsymbol{h}_t = \text{GRU}(\boldsymbol{h}_{t-1}, \text{MLP}(\text{concat}[\mathbf{z}_{t-1}, \boldsymbol{a}_t])) \,. \tag{15}$$

This is then used to compute the sufficient statistics of the prior, likelihood, and posterior:

$$p(\mathbf{z}_t \mid \mathbf{z}_{<t}, \boldsymbol{a}_{\leq t}) = \text{MLP}(\boldsymbol{h}_t) \,, \tag{16}$$

$$p(\mathbf{x}_t \mid \mathbf{z}_{\leq t}, \boldsymbol{a}_{\leq t}) = \mathcal{N}(\hat{\mathbf{x}}_t, \mathbf{1}) \,, \quad \hat{\mathbf{x}}_t = \text{Decoder}(\texttt{concat}[\boldsymbol{h}_t, \mathbf{z}_t]) \,, \tag{17}$$

$$q(\mathbf{z}_t \mid \mathbf{z}_{<t}, \boldsymbol{a}_{\leq t}, \mathbf{x}_t) = \text{MLP}(\texttt{concat}[\boldsymbol{h}_t, \boldsymbol{e}_t]) \,, \quad \boldsymbol{e}_t = \text{Encoder}(\mathbf{x}_t) \,. \tag{18}$$

The training objective is to maximize the evidence lower bound (ELBO):

$$\log p(\mathbf{x}_{0:T} \mid \boldsymbol{a}_{1:T}) \geq \mathbb{E}_q \left[ \sum_{t=0}^{T} \log p(\mathbf{x}_t \mid \mathbf{z}_{\leq t}, \boldsymbol{a}_{\leq t}) - \mathcal{L}_{\text{KL}}\Big( q(\mathbf{z}_t \mid \mathbf{z}_{<t}, \boldsymbol{a}_{\leq t}, \mathbf{x}_t), \; p(\mathbf{z}_t \mid \mathbf{z}_{<t}, \boldsymbol{a}_{\leq t}) \Big) \right] \,. \tag{19}$$

### H.2  TSSM

Our implementation of TSSM [5] uses the same generative process, approximate posterior, and training objective as S4WM. For convenience, we repeat them below. The generative process is:

$$p(\mathbf{x}_{1:T} \mid \mathbf{x}_0, \boldsymbol{a}_{1:T}) = \int p(\mathbf{z}_0 \mid \mathbf{x}_0) \prod_{t=1}^{T} p(\mathbf{x}_t \mid \mathbf{z}_{\leq t}, \boldsymbol{a}_{\leq t}) \, p(\mathbf{z}_t \mid \mathbf{z}_{<t}, \boldsymbol{a}_{\leq t}) \, \mathrm{d}\mathbf{z}_{0:T} \,, \tag{20}$$

where $\mathbf{z}_{0:T}$ are the stochastic latent states. The approximate posterior is defined as:

$$q(\mathbf{z}_{0:T} \mid \mathbf{x}_{0:T}, \boldsymbol{a}_{1:T}) = \prod_{t=0}^{T} q(\mathbf{z}_t \mid \mathbf{x}_t) \,, \quad \text{where } q(\mathbf{z}_0 \mid \mathbf{x}_0) = p(\mathbf{z}_0 \mid \mathbf{x}_0) \,. \tag{21}$$

The training objective is to maximize the evidence lower bound (ELBO):

$$\log p(\mathbf{x}_{1:T} \mid \mathbf{x}_0, \boldsymbol{a}_{1:T}) \geq \mathbb{E}_q \left[ \sum_{t=1}^{T} \log p(\mathbf{x}_t \mid \mathbf{z}_{\leq t}, \boldsymbol{a}_{\leq t}) - \mathcal{L}_{\text{KL}}\Big( q(\mathbf{z}_t \mid \mathbf{x}_t), \; p(\mathbf{z}_t \mid \mathbf{z}_{<t}, \boldsymbol{a}_{\leq t}) \Big) \right] \,. \tag{22}$$

The main difference from S4WM is that in TSSM the prior $p(\mathbf{z}_t \mid \mathbf{z}_{<t}, \boldsymbol{a}_{\leq t})$ is computed by a stack of Transformer [61] blocks. Specifically, the Transformer blocks output an embedding vector $\boldsymbol{h}_t$ through self-attention over the history:

$$\boldsymbol{h}_t = \text{TransformerBlocks}(\boldsymbol{g}_{1:t}) \,, \quad \boldsymbol{g}_t = \text{MLP}(\texttt{concat}[\mathbf{z}_{t-1}, \boldsymbol{a}_t]) \,. \tag{23}$$

The $\boldsymbol{h}_t$ is then used for predicting the next latent state $\mathbf{z}_t$ and decoding the latent state into image $\hat{\mathbf{x}}_t$:

$$p(\mathbf{z}_t \mid \mathbf{z}_{<t}, \boldsymbol{a}_{\leq t}) = \text{MLP}(\boldsymbol{h}_t) \,, \quad \hat{\mathbf{x}}_t = \text{Decoder}(\texttt{concat}[\boldsymbol{h}_t, \mathbf{z}_t]) \,. \tag{24}$$

## I  Speed and Memory Usage Evaluation Details

All results in Figure 3 are obtained on a single NVIDIA RTX A6000 GPU. We make sure that the models have comparable number of parameters. For training, we use a batch size of 8, sequence length of 1000, and image size of $64 \times 64$. We report the number of episodes per second processed by each model, averaged over 100 batches, and also the peak memory usage. For imagination, we use a batch size of 64, context length of 500, generation length of 500, and image size of $64 \times 64$. We report the number of frames per second, averaged over 8 batches.

## J  Implementation Details and Hyperparameters

We base our implementation on the publicly available code of S4 [21] and DreamerV3 [31]. We provide the hyperparameters used for 3D and 2D environments in Tables 10 and 11 respectively. For 3D environments, we largely follow the hyperparameters used in Memory Maze [50]. For 2D environments, we follow the architecture of DreamerV3-S. We use a linear warmup (1000 gradient steps) and cosine anneal learning rate schedule for S4WM. Figure 21 shows the detailed architecture of an S4 block. For TSSM-XL and RSSM-TBTT, we use constant learning rates following the original papers. To ensure a fair comparison, all models use the same CNN encoder and decoder,

with convolutional and transposed convolutional layers of kernel size 4 and stride 2, layer normalization [1], and SiLU [53] nonlinearity. The latent states $\mathbf{z}_{0:T}$ are vectors of categorical variables [30] optimized by straight-through gradients [3]. To facilitate stable training, we parameterize the categorical distributions as mixtures of $1\%$ uniform and $99\%$ neural network output [31]. We use KL balancing [30] to scale the gradient of the KL loss $\mathcal{L}_{\mathrm{KL}}(q, p)$ with respect to the posterior $q$ and prior $p$:

$$\mathcal{L}_{\mathrm{KL}}(q, p) = \alpha \cdot \mathrm{KL}[\texttt{stop\_grad}(q) \parallel p] + (1 - \alpha) \cdot \mathrm{KL}[q \parallel \texttt{stop\_grad}(p)] . \tag{25}$$

Here, $\texttt{stop\_grad}$ denotes the stop-gradient operator, and we set $\alpha = 0.8$ to put more emphasis on learning the prior toward the posterior than the other way around.

## K  Broader Impact

The proposed S4WM and other world models investigated in this paper are fundamentally deep generative models. Therefore, they inherit the potential negative social impacts that deep generative models may have, such as generating fake images and videos that can contribute to digital misinformation and deception. Caution must be exercised in the application of these models, adhering to ethical guidelines and regulations to mitigate the risks.

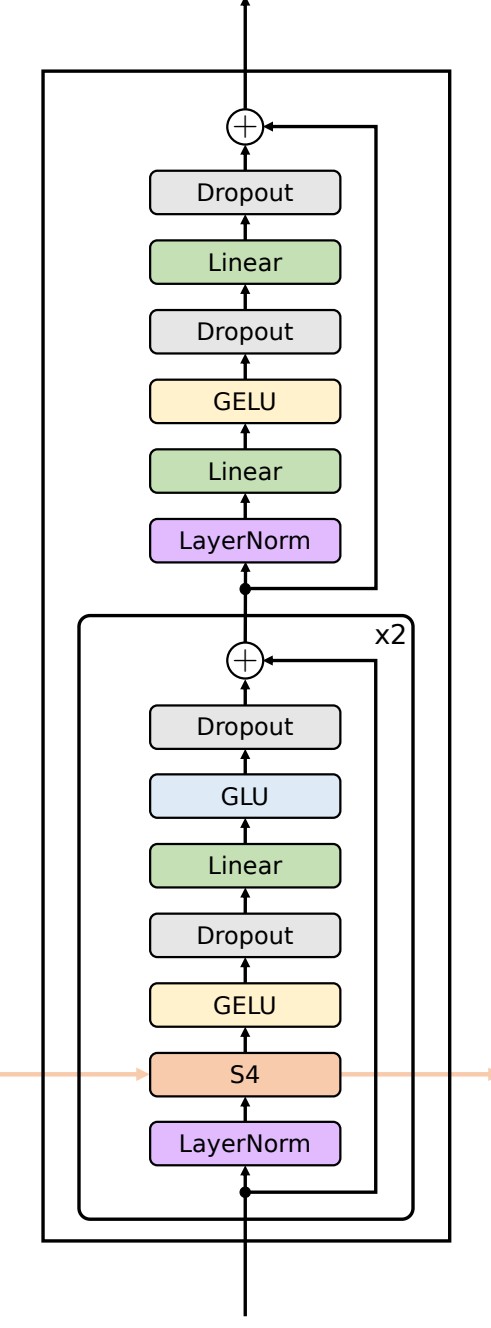

**Figure 21:** Detailed architecture of an S4 block.

**Table 10:** Hyperparameters for 3D environments.

|  | S4WM | TSSM-XL | RSSM-TBTT |
|---|---|---|---|
| **Training** | | | |
| Optimizer | AdamW | AdamW | AdamW |
| Batch size | 8 | 8 | 32 |
| Epochs | 57 | 57 | 57 |
| Learning rate | $1 \times 10^{-3}$ | $1 \times 10^{-4}$ | $3 \times 10^{-4}$ |
| Weight decay | $1 \times 10^{-2}$ | $1 \times 10^{-2}$ | $1 \times 10^{-2}$ |
| Gradient clipping | 1000 | 1000 | 200 |
| TBTT steps | – | – | 48 |
| **Model** | | | |
| Stochastic discrete latent size | $32 \times 32$ | $32 \times 32$ | $32 \times 32$ |
| History encoding blocks | 6 | 12 | 1 |
| History encoding $d_{\mathrm{model}}$ | 512 | 512 | 2048 |
| History encoding $d_{\mathrm{ff}}$ | 2048 | 512 | – |
| History encoding cache length $m$ | – | 128 / 256 / 512 | – |
| CNN layers | 4 | 4 | 4 |
| CNN multiplier | 48 | 48 | 48 |
| MLP hidden units | 1000 | 1000 | 1000 |
| Total parameters | 41.1 M | 43.8 M | 54.1 M |
| **Objective** | | | |
| KL balancing $\alpha$ | 0.8 | 0.8 | 0.8 |

**Table 11:** Hyperparameters for 2D environments.

|  | S4WM | TSSM-XL | RSSM-TBTT |
|---|---|---|---|
| **Training** | | | |
| Optimizer | AdamW | AdamW | AdamW |
| Batch size | 8 | 8 | 32 |
| Epochs | 100 | 100 | 100 |
| Learning rate | $1 \times 10^{-3}$ | $1 \times 10^{-4}$ | $3 \times 10^{-4}$ |
| Weight decay | $1 \times 10^{-2}$ | $1 \times 10^{-2}$ | $1 \times 10^{-2}$ |
| Gradient clipping | 1000 | 1000 | 200 |
| TBTT steps | – | – | 50 |
| **Model** | | | |
| Stochastic discrete latent size | $32 \times 32$ | $32 \times 32$ | $32 \times 32$ |
| History encoding blocks | 6 | 10 | 1 |
| History encoding $d_{\mathrm{model}}$ | 512 | 512 | 2048 |
| History encoding $d_{\mathrm{ff}}$ | 2048 | 512 | – |
| History encoding cache length $m$ | – | 128 | – |
| CNN layers | 3 | 3 | 3 |
| CNN multiplier | 32 | 32 | 32 |
| MLP hidden units | 512 | 512 | 512 |
| Total parameters | 28.1 M | 27.1 M | 31.2 M |
| **Objective** | | | |
| KL balancing $\alpha$ | 0.8 | 0.8 | 0.8 |

