# OpenReview forum: "Facing Off World Model Backbones: RNNs, Transformers, and S4"
_NeurIPS.cc/2023/Conference — NeurIPS 2023 poster_

### Official Review · Reviewer_kwZk · 2023-07-03

**Soundness:** 3 good
**Presentation:** 3 good
**Contribution:** 2 fair
**Rating:** 5
**Confidence:** 4

**Summary:**

This paper presents a novel architecture for world-models in deep RL, which is based on S4, a generative sequence model. Furthermore, the paper presents an evaluation to two other world-model architectures: RSSM's and transformer-based world-models.

**Strengths:**

- Integrating more powerful models into existing world-model algorithms can lead to more performant and easier to train algorithms. Hence, the general idea is valid.
- The benchmark contribution sheds some light on which models are useful as a backbone for world-model RL and is certainly beneficial for the RL community.
- The experimental evaluation is insightful and appears to be of high-quality. Especially useful are the throughput comparisons during training and inference.

**Weaknesses:**

- The idea to use novel sequence model architectures as backbones for world-models in RL is somewhat incremental.
- There is already another (more extensive) benchmark in video prediction models [(Yan et al., 2022)](https://arxiv.org/pdf/2210.02396.pdf). While this study does not explicitly target world-models, it contains models that have been applied as such. Furthermore, those models are also action-conditioned, which means they should be able to serve as world-models as well. This paper misses this, quite important, reference.
- The benchmark lacks some depth and detail. While the general setup, environment selection and metrics are suitable, it would be great to have some more direct insight. For instance, show also the path that was taken during trajectory reconstruction. This would make it easier to understand how the agent's path influences the prediction. Also, it would give some insight on how the reconstruction behaves on loop-closures (revisiting old locations) and identical looking positions. In general, different types of runs (loops, straight lines, etc) would benefit the evaluation.
- The approach is only tested only on offline data. While the rationale behind it is clear (comparing w.r.t model quality only), i think it would still make sense to use it actually as a world model for RL. I acknowledge that training world-model RL algorithms is not easy and can be brittle. However, it not necessary to present SOTA results for some world in this regard, just to have some evidence and insight on how well it actually does in the task for this model was developed.

**Questions:**

- I am willing to reconsider my evaluation score if there is some discussion of the included reference (Yan et al., 2022) and how this study sets itself apart.
- Equation 5: This equation states that $h_{t-1}$ is fed into the S4 block. Figure 1. however does not show this.

**Limitations:**

The limitation of the benchmark environment simplicity was acknowledged.

---

> ### Author Response · Authors · 2023-08-11
> **Response to Reviewer kwZk**
>
> We sincerely appreciate your valuable insights! We hope we have addressed your concerns about incremental contribution, discussion about TECO, and evaluation with MBRL agent in our "global" response. We address your remaining concerns below. Please let us know if you have additional comments.
>
> > Visualization of agent paths; different types of runs (loops, straight lines, etc)
>
> Thank you for your suggestion. We have included videos in the supplementary material showing typical agent paths and the imagination along the way. As can be seen from the video, the agent's paths in the 3D environment are mostly loops, while in the 2D Distracting Memory environment, the agent moves in straight lines.
>
> > Inconsistency between Equation 5 and Figure 1
>
> Thank you for pointing this out. We will fix it in the revision.

---

> > ### Comment · Reviewer_kwZk · 2023-08-14
> >
> > I thank the authors for their response.
> > Due to the added discussion on Yan et al. (2022), I am willing to raise my score from 4 to 5.

---

### Official Review · Reviewer_vwzx · 2023-07-03

**Soundness:** 3 good
**Presentation:** 2 fair
**Contribution:** 2 fair
**Rating:** 5
**Confidence:** 4

**Summary:**

Model-Based RL algorithms rely on learning efficient world models for learning optimal behaviors. Existing agents like Dreamer rely on an RNN for learning world models which have limited memory capacity. S4 has shown promising results when compared with RNNs and Transformers in terms of both complexity and efficiency on various tasks. This work proposes a method called S4WM that uses S4 as the backbone of world models. The method is proposed for high-dimensional inputs (images) under a probabilistic learning framework and uses variational inference for learning. The architecture seems similar to Dreamer, where the deterministic component (RNN) in Dreamer is replaced with a stack of S4 layers. The paper also proposes a few benchmarks for evaluation in an offline learning setting and shows that S4WM outperforms RNN and Transformers-based baselines.

**Strengths:**

1. The problem statement of improving the memory capacity of world models while being computationally efficient by deploying S4 layers is interesting.
2. The experiments thoroughly study different aspects to test the ability of proposed methods on tasks like reward prediction, memory, and reasoning.

**Weaknesses:**

1. Although the paper introduces new benchmarks, it is important to evaluate methods on the existing benchmarks (Questions 3, 4).
2. Some of the baselines and related works are missing in the paper (Question 2) which focused on a similar problem of improving the long-term memory of world models.
3. Section 4 is not well explained. The paper should highlight the differences with the Dreamer architecture. From my understanding, S4 layers are used to replace the GRU in the deterministic component.

**Questions:**

1. Line 171 in the paper mentions that- “ We make sure that the models have comparable number of parameters.”. What were the hyperparameters used for RSSM/TSSM-XL/S4WM to have a similar number of parameters? Was the same number of hyperparameters only used to run the validation for memory usage and speed (in Fig 3) or for other experiments too?
2. There have been more recent works on improving RSSM to have better memory capacity- Clockwork VAEs [1] and VSG [2]. How well does S4WM compare with such methods? Clockwork VAEs did not do the action-conditioned generation of frames but should be mentioned in related works, however, they are used in [4] for evaluation.
3. Although the paper proposed benchmarks for evaluation, the methods should be evaluated on offline datasets used in [3]. Section 5 discusses some offline experiments and uses probing for evaluation. Rather than using MSE as the evaluation metric, it would be interesting to observe the results of probing to test the quality of learned embeddings. Is there a way to do probing for the benchmarks used in the paper?
4. TECO [4] is another baseline that uses Transformers for action-conditioned video generation and reported results on benchmarks like Habitat and Minecraft, and should be added for comparison.
5. How do these methods compare when only a few initial frames (maybe 20) are provided and open-loop prediction is done conditioned on them?
6. Is there a way to do some kind of probing to evaluate the embeddings of the world models rather than using MSE?
7. The details of the S4 block are not mentioned in the paper. If there are multiple layers, was an experiment conducted to study the effect of the number of layers in the S4 block?
 \
### References
[1] Saxena, Vaibhav, Jimmy Ba, and Danijar Hafner. "Clockwork variational autoencoders." Advances in Neural Information Processing Systems 34 (2021): 29246-29257.\
[2] Jain, Arnav Kumar, et al. "Learning robust dynamics through variational sparse gating." Advances in Neural Information Processing Systems 35 (2022): 1612-1626.\
[3] Pasukonis, Jurgis, Timothy Lillicrap, and Danijar Hafner. "Evaluating Long-Term Memory in 3D Mazes." arXiv preprint arXiv:2210.13383 (2022).\
[4] Yan, Wilson, et al. "Temporally consistent video transformer for long-term video prediction." arXiv preprint arXiv:2210.02396 (2022).

**Limitations:**

The only limitation mentioned in the paper is to extend it to more complex environments. More limitations of the current algorithm should be discussed. For example, the paper needs to discuss how to extend this to online learning and if there will be any bottlenecks when moving to harder tasks or stochastic environments.

---

> ### Author Response · Authors · 2023-08-11
> **Response to Reviewer vwzx**
>
> We sincerely thank you for your constructive feedback! We hope we have addressed your concerns about baseline comparison in our "global" response. S4WM is significantly better than Clockwork VAE on DMLab in the TECO benchmark. S4WM is also significantly better than TECO in our Two Rooms environment when both models are trained for 80K steps. We will also expand the Related Work section. We address your remaining concerns below. Please let us know if you have additional comments.
>
> > Hyperparameters
>
> We list the hyperparameters and model size in supplementary material Table 4 and 5. To obtain the speed and memory usage in Figure 3, we use the same model hyperparameters in supplementary material Table 4, which are also used for our experiments in 3D environments.
>
> > Offline probing
>
> Thank you for your suggestion. We provide offline probing results on the Memory Maze benchmark in supplementary material Section C Table 3. For all three models, we use $\texttt{concat}[h_t, z_t]$ as the input to the probing network. Both TSSM and S4WM underperform RSSM. We conjecture that this is because for TSSM and S4WM, $\texttt{concat}[h_t, z_t]$ is not the optimal embedding for probing. We will investigate other embedding choices such as hidden states from intermediate layers.
>
> > Open-loop prediction with only a few initial frames (maybe 20)
>
> We note that with only a few initial frames, the agent can only see a very small portion of the environment. Since we evaluate the generation quality in hold-out environments, it is impossible for the agent to correctly imagine the rest of the environment based on only a few initial frames. Please refer to the supplementary videos for typical agent paths in the environments.
>
> > Details of the S4 block
>
> Thank you for your suggestion. We will add more details in the revision. The structure of the S4 block is similar to what is used in previous work. We set the number of layers to ensure that S4WM has comparable number of parameters as TSSM and RSSM.
>
> > Highlight the differences with the Dreamer architecture
>
> Thank you for your suggestion. We provide a brief description of RSSM and TSSM in supplementary material Section D. In addition to replacing the GRU with S4 blocks, the posterior design is also different. In RSSM, the posterior takes the GRU hidden state as input, while in S4WM, the posterior is computed directly from the input image (or optionally through another set of S4 blocks, see supplementary material Section B.1 for details). This design ensures the parallel trainability of S4WM.

---

> > ### Comment · Reviewer_vwzx · 2023-08-17
> >
> > I thank the authors for the clarifications to my concerns. I have a few more questions regarding the Offline Probing part:
> > 1. For the concat operation, was $h_t$ for S4WM obtained by concatenating the output from all the S4 blocks?
> > 2. Is the method not able to learn when the maze layouts are changing?
> > 3. Does the paper have an experiment on probing where the maze layout is fixed? This would tell if the point 2 is correct or not.

---

> ### Author Response · Authors · 2023-08-21
> **Response to Reviewer vwzx**
>
> Thank you for your additional comments! For the offline probing experiments in the supplementary material, $h_t$ is the output of the final S4 block. Following your suggestions, we conducted the probing experiments in our Four Rooms environment, where the maze layout is fixed but the object positions are random. We use $\texttt{concat}[h_t, z_t]$ to predict the object positions, as similarly done in the Memory Maze benchmark. In addition, we tried concatenating outputs from all S4 blocks (denoted S4WM-All-Blocks). We report the prediction MSE below (lower is better). RSSM still performs the best. We conjecture that this is because in RSSM $\texttt{concat}[h_t, z_t]$ captures more global information while in TSSM and S4WM $\texttt{concat}[h_t, z_t]$ captures more local information for each $t$.
>
> | RSSM-TBTT | TSSM-XL | S4WM | S4WM-All-Blocks |
> |:---------:|:-------:|:----:|:---------------:|
> |   2.05    |  3.19   | 2.85 |      2.58       |
>
> We emphasize that we are not proposing S4WM as the best world model in all aspects. Instead, we contribute a thorough investigation to understand the pros and cons of different world model backbones. We believe that even if S4WM is inferior in some aspect, the new knowledge is worth sharing with the community.

---

> > ### Comment · Reviewer_vwzx · 2023-08-21
> >
> > Thank you for conducting the experiment in such a short time. I hope the authors add these results in Limitations or Discussion as neither TSSM nor S4WM do not outperform RSSM (which is a simple model with a GRU). I will update my score.

---

### Official Review · Reviewer_TTTx · 2023-07-04

**Soundness:** 4 excellent
**Presentation:** 4 excellent
**Contribution:** 3 good
**Rating:** 6
**Confidence:** 5

**Summary:**

This paper studies the backbones of world models. An S4-based world model S4WM is proposed by replacing RNNs in RSSM with S4 Blocks, which is known to be good at long-term memory. An empirical study has been conducted to compare three major backbones, RNN, Transformers, and S4, on speed, memory usage, and performance of multiple memory-demanding tasks.

**Strengths:**

1. Introduction of S4 into the field of world models and MBRL is appreciated
2. Carefully designed memory-demanding tasks and extensive experiments
3. This paper is well-written and easy to follow

**Weaknesses:**

1. Eq. (6), in order to obtain parallel trainability, approximates posterior only based on the current observation, regardless of previous latent states. This approximation, as pointed out in the paper of TSSM, is myopic and may hurt memory ability, despite satisfactory empirical performance in evaluated environments.
2. This work does not evaluate MBRL equipped with S4WM on memory-demanding tasks, such as DMLab.

**Questions:**

Only one minor question:

1. The large memory usage of RSSM for imagination (Fig 3d) is attributed to the image decoder. Why does not the decoder also make a high memory usage in RSSM for training (Fig 3c)?

**Limitations:**

This work has discussed its limitations and future work in the conclusion. I recommend discussing weaknesses 1 and 2 mentioned above in the part on limitations.

There does not seem to be any negative social impact of this paper that should be discussed.

---

> ### Comment · Reviewer_TTTx · 2023-08-11
>
> I thank the author for their response. I appreciate the effort made on the preliminary evaluation of S4WM for MBRL (planning).
>
> However, I respectfully ask for a further discussion on Weakness 1 above.

---

> ### Author Response · Authors · 2023-08-11
> **Response to Reviewer TTTx**
>
> We would like to express our appreciation to your positive recommendation! We hope the skill-level MPC experiment shows the potential of equipping MBRL agents with S4WM. We answer your remaining questions below. Please let us know if you have additional comments.
>
> > Myopic approximate posterior for parallel trainability
>
> We show in the supplementary material Section B.1 that it is possible to condition the posterior on the full history and maintain the ability to train in parallel. We call this model S4WM-Full-Posterior. We also provide a full comparison between S4WM and S4WM-Full-Posterior in terms of generation quality, speed, and memory consumption. In general, S4WM-Full-Posterior can obtain better generation quality at the cost of more computation.
>
> > Low memory usage in RSSM training
>
> This is because RSSM only encodes a chunk of length ~50 during training, while S4WM and TSSM encode the full sequence in parallel.

---

> > ### Comment · Reviewer_TTTx · 2023-08-11
> >
> > Thanks for the additional response. This has addressed my remaining issues.
> >
> > I have also read other reviews and responses. Considering preliminary experiments on MBRL and comparison with the recent TECO baseline, I decided to maintain my rating as a weak positive recommendation. I wish the authors good luck with the submission.

---

### Official Review · Reviewer_ALhP · 2023-07-06

**Soundness:** 3 good
**Presentation:** 3 good
**Contribution:** 2 fair
**Rating:** 4
**Confidence:** 3

**Summary:**

This paper introduces S4WM, the first world model based on Structured State Space Sequence (S4) models for MBRL agents. Specifically, the proposed world model leverages a stack of S4 blocks for temporal prediction in latent space. CNN encoder and decoder are used to map the image to the latent space. Compared to RNNs and Transformers, S4WM demonstrates superior long-term memory.

**Strengths:**

- The method extends the application of S4 in world models, and demonstrates better performance than RNN and transformer-based world models, especially in capturing long-term memory.
- The authors make full comparisons among RNNs, Transformers, and S4WMs in different environments.

**Weaknesses:**

- The contribution of the proposed backbone is incremental as it heavily relies on S4.
- The efficiency of the proposed method in the imagination process is worse than that of the widely used RSSM, which will bring additional overhead for MBRL agents.
- As the proposed method aims to provide a better world model for MBRL agents, a natural question is how the  ability to capture long-term dependency can benefit MBRL agents. The authors may evaluate  advanced MBRL algorithms with your improved world model.
- The experiments are all conducted in visually simple environments. I am curious whether the proposed method can handle more complex dynamics, such as robot manipulation.

**Questions:**

- As mentioned in the weakness section, the impact of the improved world model on MBRL algorithms.
- It seems that during training, the model only performs one-step prediction. I wonder whether multi-step prediction during training can further benefit the generation process.

**Limitations:**

The limitation stated is the lack of evaluation in more complex and stochastic environments.

---

> ### Author Response · Authors · 2023-08-11
> **Response to Reviewer ALhP**
>
> We sincerely appreciate your valuable feedback! We hope we have addressed your concerns about incremental contribution and evaluation with MBRL agent in our "global" response. We address your remaining concerns below. Please let us know if you have additional comments.
>
> > Lower imagination throughput than RSSM
>
> We agree and acknowledge that RSSM and S4WM have complementary strengths. As we have shown in the experiments, while RSSM imagination is faster, it fails to remember long-term history and generate consistent rollouts. This leads to lower task success rate when the agent uses the world model for planning, as shown in the skill-level MPC experiment in our "global" response. Our study presents to the community a deeper understanding of the pros and cons of different world model backbones. We consider it an interesting future direction to further improve the imagination efficiency of S4WM.
>
> > Visually simple environments
>
> In this work, we focus on modeling agent trajectories with long-range dependencies. Although the environments are visually simple, they still present significant challenges involving long-term memory. Specifically, we evaluate the world models on extremely long sequences of up to 2000 steps, and we have shown that S4WM has a clear advantage in terms of memory capacity.
>
> We agree that it is an important direction to tackle visually complex environments. However, this will likely require larger models and more complex encoder/decoder design, which becomes computationally expensive when coupled with extremely long sequences. Therefore, we would like to consider it in future work.
>
> > One-step vs multi-step prediction during training
>
> The RSSM paper proposed latent overshooting as a principled way to do multi-step prediction during training. However, latent overshooting was not necessary for RSSM to work well, and therefore it was not used in the Dreamer agents. In the context of S4WM, a naive implementation of latent overshooting will break the parallel trainability. Nevertheless, we agree that it would be interesting to see the effect of latent overshooting and will investigate it further.

---

> > ### Comment · Reviewer_ALhP · 2023-08-21
> >
> > I appreciate the author's rebuttal, which addresses some of my concerns. The paper builds the world model based on S4 and achieves good performance.
> > However, the limitations still remain, and it would be better if the method can be evaluated more thoroughly with MBRL methods in more complex environments. I think this is a borderline paper and will keep the score as-is.

---

### Official Review · Reviewer_hTvu · 2023-07-10

**Soundness:** 3 good
**Presentation:** 3 good
**Contribution:** 2 fair
**Rating:** 5
**Confidence:** 4

**Summary:**

This work aims to better model based RL agents by focusing on developing robust world models. They propose S4WM which uses the S4 model as backbone and incorporates actions as context in the hidden state. They evaluate on prediction tasks for image sequences through latent imagination and show experimental results to compare with other common sequence methods such as RNNs and Transformers.

**Strengths:**

* easy application of vanilla S4 SSM without much modification, results appear well in terms of matching colors and scenes in the Four Rooms environment
* it recognizes and seeks to address the issue of conditional generation of sequences, which is a current limitation of S4 formulated SSMs and an interesting direction to consider
* results shown make sense, showing higher throughout from recurrent model and memory efficiency for the SSM / Transformer based models
* paper well-written with few typos and simple summary of related works

**Weaknesses:**

* Would be good to add to background section for those less familiar with world model learning the common techniques and current state of progress in terms of long term memory
* I'm also curious why we only do action conditioning -- would it be effective to also evaluate the effects of conditioning on rewards / hallucinated rewards given the access to the emulator in this task setting?
* This work seems to be a direct application of the S4 framework in the MBRL domain with few comparisons to other SOTA MBRL methods. I also wonder if there needs to be different approaches for conditioning, perhaps a hybrid attention based mechanism instead of direct concatenation and mixing, which has been common in some of the recent conditional diffusion architectures, in order to improve the outcomes of latent space conditioning?
* It seems as well that as the context length is increase, the trend of MSE across the three models appear more consistent with each other. Is there a threshold where one would catch up and/or overtake the other? If so, can this be another limitation of the method themselves at scale?
* Are there additional baselines for showing the two / four / ten rooms and how they scale w.r.t. the same combinations of query / context steps? Could be important in ensuring that the results observed are robust and validated in a gradual manner.

**Questions:**

* why did the author choose the games that were used for evaluation? Are there potential nuances in stochastic vs non-stochastic dynamics that would also affect how easily the method can generalize?
* in terms of long range dependency modeling, why would action-dependent modeling affect the ability to evaluate the memorization capabilities of a model?
* Is there a set of synthetic tasks that may be a sufficient indicator for the MBRL problem itself?
* In Figure 4, what makes it harder for other models to reconstruct the correct color information? Is that a latent variable that is difficult to store efficiently in memory?
* Is it possible to set query steps beyond the context steps to check for some notion of generalization or circularity? Could be interesting to investigate the inductive biases of the recurrence versus other non-recurrent modules.
* Does good recall in the two rooms correlate to good recall in the four and proportionally in the ten as well?

**Limitations:**

Not applicable.

---

> ### Author Response · Authors · 2023-08-11
> **Response to Reviewer hTvu (1/2)**
>
> We sincerely thank you for your positive feedback! We hope we have addressed your concerns about baseline comparison in our "global" response. S4WM is significantly better than Clockwork VAE on DMLab in the TECO benchmark. S4WM is also significantly better than TECO in our Two Rooms environment when both models are trained for 80K steps. We will include more results in the Four/Ten Rooms environments. We address your remaining concerns below. Please let us know if you have additional comments.
>
> > Add background about long term memory in world models
>
> We appreciate your suggestion and will incorporate it in our revision.
>
> > Conditioning on rewards / hallucinated rewards
>
> We acknowledge that reward conditioning has been found useful in Decision Transformer (a model-free approach). However, in model-based RL, it has been common practice to only do action conditioning. This is mainly because the goal of world model learning is to approximate the state transition $p(s_{t+1}|s_t, a_t)$, which is part of the definition of Markov Decision Processes and is independent of the reward. It is expected that the state $s_t$ captures all information from the history (including rewards received up to time $t$) that is relevant to future prediction.
>
> > Other approaches for action conditioning
>
> Each action $a_t$ is simply a one-hot vector, unlike in text-to-image models where the text condition is a sequence of word embeddings. Therefore, we follow the common practice in world model learning and directly concatenate the action with the latent state. We use the same strategy for all three models for fair comparison.
>
> > Is there a threshold on sequence length where one model would catch up and/or overtake the other?
>
> For each environment, we provide three versions with short, medium, and long sequence lengths. In general, the model performance decreases with longer sequences. We find that S4WM is almost always the best across all sequence lengths (up to 2000 steps), and that at the sequence length of 1000, only S4WM can still make consistent generation. These results suggest that the performance of S4WM decays the slowest (up to sequence length of 2000). We have also shown that all models struggle at the sequence length of 2000 (which is indeed a limitation of these models), and therefore comparing model performance on even longer sequences becomes less important.

---

> ### Author Response · Authors · 2023-08-11
> **Response to Reviewer hTvu (2/2)**
>
> > Stochastic vs non-stochastic dynamics
>
> We choose non-stochastic environments mainly because there exists a single groundtruth future sequence for evaluating the generation consistency. In stochastic environments, there are multiple possible futures at each time step, causing the total number of correct future sequences to grow exponentially with the prediction length. This makes evaluation of generation consistency much harder. We believe it is worth future investigation to find proper ways to evaluate generation consistency in stochastic environments.
>
> > Why would action-dependent modeling affect the ability to evaluate the memorization capabilities of a model?
>
> First, we focus on world models which aim to approximate the state transition $p(s_{t+1}|s_t, a_t)$ and therefore have to be action-conditioned. Second, action conditioning makes it easier to obtain non-stochastic transitions, and in turn simplifies evaluation.
>
> > Is there a set of synthetic tasks that may be a sufficient indicator for the MBRL problem itself?
>
> We don't have a clear answer yet. There are quite a few desired abilities of MBRL agents, such as exploration, long-horizon planning, skill learning, robustness to distractions, etc. It is extremely challenging for an MBRL agent to possess all these abilities. Therefore, multiple benchmarks have been developed, each focusing on one or several aspects. In this work, we focus on long-term memory capacity with extremely long sequences of up to 2000 steps.
>
> > What makes it harder for other models to reconstruct the correct color information?
>
> To predict the correct color, the model needs to remember what it has seen hundreds of time steps ago (please refer to the supplementary videos for typical agent paths). RSSM and TSSM fail mainly because they have limited memory capacity. RSSM is based on RNN, which is prone to the vanishing gradients problem, making it hard to store information over long time horizons. TSSM is based on Transformer, which has quadratic complexity with respect to its context window length (denoted $m$ in the paper). In practice, we need to limit $m$ for computational efficiency, and it will be hard to propagate information for much longer than $m$ steps.
>
> > Set query steps beyond the context steps
>
> Thank you for your suggestion. We note that in the Multi Doors Keys environment, we have used query steps that are more than twice the context steps (Table 3). We find that S4WM significantly outperforms RSSM and TSSM in this setting. To further answer your question, we provide the generation result in the Two Rooms environment with 300 context steps and 700 query steps. Please refer to Figure 2 in the uploaded pdf. We note that during training, the model has only seen sequences of length 500 (300 context + 200 query). Our results show that S4WM can successfully generalize to sequence lengths that are beyond the training regime.
>
> > Does good recall in the two rooms correlate to good recall in the four and proportionally in the ten as well?
>
> We don't think there is a correlation. Good recall becomes more challenging when there are more rooms, as the model needs to remember more information. We use two/four/ten rooms as three difficulty levels to thoroughly compare the memory capacity of different models.

---

### Author Rebuttal · Authors · 2023-08-10

We thank all reviewers for their insightful and constructive feedback. We are encouraged that they appreciate our contribution of introducing S4 into the field of world models to improve long-term memory capacity (@TTTx, @vwzx). Many reviewers praised our thorough and high-quality experiments that evaluate several aspects of world models across different backbones and environments (@ALhP, @TTTx, @vwzx, @kwZk), which can be beneficial for the RL community. We address the main concerns from reviewers below.

> Evaluate MBRL equipped with S4WM (@ALhP, @ TTTx, @ kwZk)

We appreciate the suggestion. With S4WM, we aim to make a step towards tackling environments with long-range dependencies. Such environments entangle several challenges such as exploration, long-horizon planning and skill learning in addition to the world model learning. An agent with any of these pieces missing may fail in such environments. Therefore, successfully training an MBRL agent from scratch in such environments is extremely difficult, even if the agent is equipped with better world models such as S4WM.

To address the reviewer concern, we instead consider a skill-level Model-Predictive Control (MPC) agent, with predefined skills and an offline-trained world model. This will show whether the world model can benefit RL tasks through planning. Specifically, we consider the Multi Doors Keys environments. The predefined skills are (1) picking up a key at a specific position, and (2) going to a door at a specific position and attempting to unlock it. The agent will return to its starting position at the end of each skill. The task is to unlock a door specified by a goal image, which shows the door in unlocked state. The agent is allowed to execute two skills: pick up one key and then attempt to unlock one door. Hence, planning has two stages. First, the agent enumerates all allowed skill sequences (of length 2), uses the world model to predict which skill sequence is most likely to unlock the correct door. Then the agent executes the first skill in the environment and replan. We show the success rate of RSSM, TSSM, and S4WM in Table 1 of upload pdf. The MPC agent equipped with S4WM is able to unlock every door in every environment, while RSSM and TSSM succeed roughly half of the time.

> Compare with additional baselines such as TECO (@hTvu, @vwzx, @kwZk)

We thank reviewers for pointing to this relevant paper, which is published at ICML 2023. We note that TECO considers more visually complex environments, with sequence length capped at 300. Therefore, TECO uses the quite heavy VQGAN encoder/decoder, and a large model of 169M parameters. In contrast, we consider extremely long sequences (up to 2000 steps), where using heavy models becomes too costly. For this reason, we mainly considered visually simpler environments that can be tackled with lightweight models (S4WM has 41M parameters).

Following the reviewer suggestions, we provide a quantitative comparison with additional baselines in Table 2 of uploaded pdf. This comparison is done on DMLab following the evaluation protocol of the TECO paper. All baselines use a pretrained VQGAN encoder/decoder, while S4WM uses a jointly trained simple CNN encoder/decoder (we use the same hyperparameter as our 3D experiments). S4WM significantly outperforms Clockwork VAE and Latent FDM (a diffusion-based model), and is close to TECO.

We further showcase the training efficiency of S4WM in Table 3 of uploaded pdf. Here we train both S4WM and TECO for only 80K steps in the Two Rooms environment (in Table 2, the models are trained for 1M steps). We find that S4WM performs well while TECO fails to learn as quickly as S4WM. This suggests that S4WM may better suit online MBRL agents, where the world model needs to constantly adapt to the data collected by the agent.

> The contribution of combining S4 with world models is somewhat incremental (@ALhP, @kwZk)

We acknowledge that the proposed model S4WM is not completely new, but a novel combination of existing components. In addition to proposing S4WM as the first S4-based world model, we also contribute an extensive empirical study of the three major world model backbones in terms of several important memory capabilities evaluated in carefully designed environments. We believe that this thorough investigation presents new knowledge and can be valuable to the community.

---

### Decision · Program_Chairs · 2023-09-21

**Decision:**

Accept (poster)

**Comment:**

This paper proposes to use S4 as a backbone for world models. Authors compare RNN, Transformers, and S4 on various prediction tasks based on the learnt world model and show that S4 is superior to both RNNs and Transformers.

The authors have addressed most of the reviewers' concerns during the rebuttal. There was a question of applying the proposed world model to MBRL. The authors added some MPC results in the rebuttal. However, I don't think it is very convincing application of the proposed world model. Hence, I suggest the authors make it explicit in the abstract that this is not a paper about using S4 for MBRL. It is only about using S4 for supervised learning of the world model.